# Increased reluctant vesicles underlie synaptic depression by GPR55 in axon terminals of rat cerebellar Purkinje cells

**Takuma Inoshita, Shin-ya Kawaguchi\***

Department of Biophysics, Graduate School of Science, Kyoto University Oiwake-cho, Kitashirakawa, Sakyo-ku, Kyoto, Japan

## eLife Assessment

This is an **important** study reporting that activation of the presynaptic GPR55 receptor suppresses synaptic transmission by modulating GABA release through the reduction of the readily releasable pool without affecting the presynaptic AP waveform and calcium influx. The evidence supporting this claim is **compelling** and based on an impressive array of techniques including patch-clamp recordings from the axon terminals of cerebellar Purkinje cells and fluorescent imaging of vesicular exocytosis. While the authors have strengthened their conclusions on several technical fronts in the revised version, further investigation is needed into the mechanism by which GPR55 activation might make vesicles insensitive to the rise in presynaptic [$Ca^{2+}$] mediated by VGCCs, and the nature of the endogenous process that would activate this pathway in vivo.

**\*For correspondence:**
kawaguchi.shinya.7m@kyoto-u.ac.jp

**Competing interest:** The authors declare that no competing interests exist.

**Abstract** Control of synaptic transmission efficacy by neuronal activity and neuromodulators is pivotal for brain function. Synaptic suppression by cannabinoids activating CB1 receptors has been extensively studied at the molecular and cellular levels to understand the neuronal basis for effects of cannabis intake. Here, we focused on GPR55, a non-canonical type of cannabinoid receptor, which shows sensitivity to cannabidiol included in cannabis, aiming to highlight its actions on presynaptic function. Taking advantage of direct patch-clamp recordings from axon terminals of rat cerebellar Purkinje cells together with fluorescent imaging of vesicular exocytosis using synapto-pHluorin, we show that GPR55 suppresses synaptic transmission as CB1 receptor does, but through a distinct presynaptic modulation of release machinery. Activation of GPR55 reduced transmitter release by changing neither presynaptic action potential waveform nor $Ca^{2+}$ influx, but by making a large population of $Ca^{2+}$-responsive synaptic vesicles insensitive to $Ca^{2+}$ influx through voltage-gated $Ca^{2+}$ channels, leading to substantial reduction of the readily releasable pool of vesicles. Thus, the present study identifies a unique mechanism to suppress presynaptic transmitter release by an atypical cannabinoid receptor GPR55, which would enable subtype-specific modulation of neuronal computation by cannabinoid receptors.

## Introduction

Neuronal communication at chemical synapses is central to information processing in the circuit, and its activity-dependent modulation provides adaptive change of brain computation (**Kandel, 2001**). Short- and long-term changes of transmission efficacy at central synapses have been extensively studied in terms of molecular/cellular mechanisms and physiological significance, which have demonstrated their critical roles in learning and memory in animals (**Abbott and Regehr, 2004**). One of such modulations of synaptic strength is that by endocannabinoids (**Kreitzer and Regehr, 2001**;

*Ohno-Shosaku et al., 2001*; *Wilson and Nicoll, 2001*; *Diana et al., 2002*), which is also related to the cellular and neuronal basis for actions of cannabis (*Kano et al., 2009*). One well-known synaptic modulation by cannabinoids is the activity-dependent short-term suppression of synaptic transmission, such as depolarization-induced suppression of excitation (DSE) and inhibition (DSI), caused by reduced transmitter release from presynaptic terminals as a result of activation of cannabinoid receptor type 1 (CB1). The CB1 receptor activation is thought to decrease presynaptic $Ca^{2+}$ influx upon an action potential (AP) arrival (*Kreitzer and Regehr, 2001*; *Kushmerick et al., 2004*; *Wu et al., 2020*).

In spite of the substantial efforts that have been made for unraveling cannabinoids' action on the nervous system, it still remains elusive whether the reduction of $Ca^{2+}$ influx fully explains the suppression of transmission by cannabinoids or other mechanisms, such as altered release machinery, may also be involved generally at central synapses, as already suggested for CB1-mediated change of presynaptic vesicular docking (*Ramírez-Franco et al., 2014*). In addition, multiple types of molecules contained in cannabis are now known to act on the nervous system, and particularly one of them, cannabidiol, is attracting attention because of its potential preferred therapeutic actions on various neurological dysfunctions such as epilepsy and anxiety symptoms (*Shallcross et al., 2019*; *Devinsky et al., 2024*). GPR55, a cannabinoid receptor that has been identified as a potential target of cannabidiol (*Baker et al., 2006*; *Oka et al., 2007*), was recently reported to modulate synaptic transmission at glutamatergic and GABAergic synapses in the hippocampus (*Sylantyev et al., 2013*; *Rosenberg et al., 2023*), although the detailed mechanism of action remains unclear. GPR55, unlike typical cannabinoid receptors such as CB1 receptors that couple to $G_i$ proteins, interacts with multiple G proteins including $G\alpha_q$ and $G\alpha_{13}$, thereby activating downstream pathways such as phospholipase C (PLC)/ $IP_3$-mediated $Ca^{2+}$ signaling and RhoA/ROCK-dependent cytoskeletal modulation (*Henstridge et al., 2008*). This distinct signaling profile suggests that GPR55 is likely to regulate synaptic transmission through a mechanism different from CB1 receptor.

To study in detail how cannabinoids modulate synaptic physiology, direct *tour-de-force* patch-clamp recording of transmitter release at axon terminals, in combination with molecular biological methods and fluorescent imaging, would be a powerful technique. One preparation that is amenable to detailed biophysical analysis of transmitter release is the axon terminals of cerebellar Purkinje cells (PCs) (*Kawaguchi and Sakaba, 2015*), which are expected to express GPR55 based on data from in situ hybridization, RT-PCR, and immunolabeling (*Ryberg et al., 2007*; *Wu et al., 2013*), although its subcellular localization remains unclear. Taking advantage of the feasibility to perform direct patch-clamp recording from axon terminals in combination with fluorescent imaging of vesicular fusion in dissociated cultures and postsynaptic recordings from neurons in deep cerebellar nuclei (DCN) of slices, here we studied how the presynaptic release machinery is modulated by GPR55. Here, we demonstrate that GPR55 is present at PC axon terminals and suppresses synaptic transmission through a unique modulatory action: GPR55 increases reluctant vesicles by depriving the sensitivity to APs.

## Results

### GPR55 inhibits synaptic transmission at PC output synapses

We first examined the role of GPR55 in synaptic transmission from a PC in acute cerebellar slices. Cerebellar sagittal slices were prepared from postnatal day (P)28–35 rats, and whole-cell patch-clamp recordings were performed from neurons with large cell bodies (>15 μm diameter) in the DCN (*Figure 1A*; *Uusisaari et al., 2007*). The DCN neurons were voltage-clamped at –70 to –100 mV, and evoked IPSCs (eIPSCs) were recorded following electrical stimulation at the white matter in the presence of an AMPA receptor antagonist, NBQX (10 μM). In accord with previous studies (*Kawaguchi and Sakaba, 2015*), eIPSCs at PC-DCN synapses were identified by the sudden appearance of large amplitude responses (2.04±0.48 nA) in an almost all-or-none manner as stimulation intensity was increased (*Figure 1B*). Bath application of a GPR55 agonist, AM251 (5 μM; $EC_{50}$=39 nM assayed in HEK cells; *Ryberg et al., 2007*; *Henstridge et al., 2008*) decreased the amplitude of eIPSCs (69.1 ± 5.1% at +5 min, p=0.00226, compared with –1 min, paired *t*-test), accompanied by an increase in the coefficient of variation (CV) of the responses (156.0 ± 19.8% at +5 min, p=0.0490) (*Figure 1B and C*). An increase in the CV is typically interpreted as a lowered presynaptic release probability. These results suggest that GPR55 suppresses synaptic transmission at PC-DCN synapses, similarly to what CB1 receptor does at other synapses (*Kreitzer and Regehr, 2001*; *Ohno-Shosaku et al., 2001*;

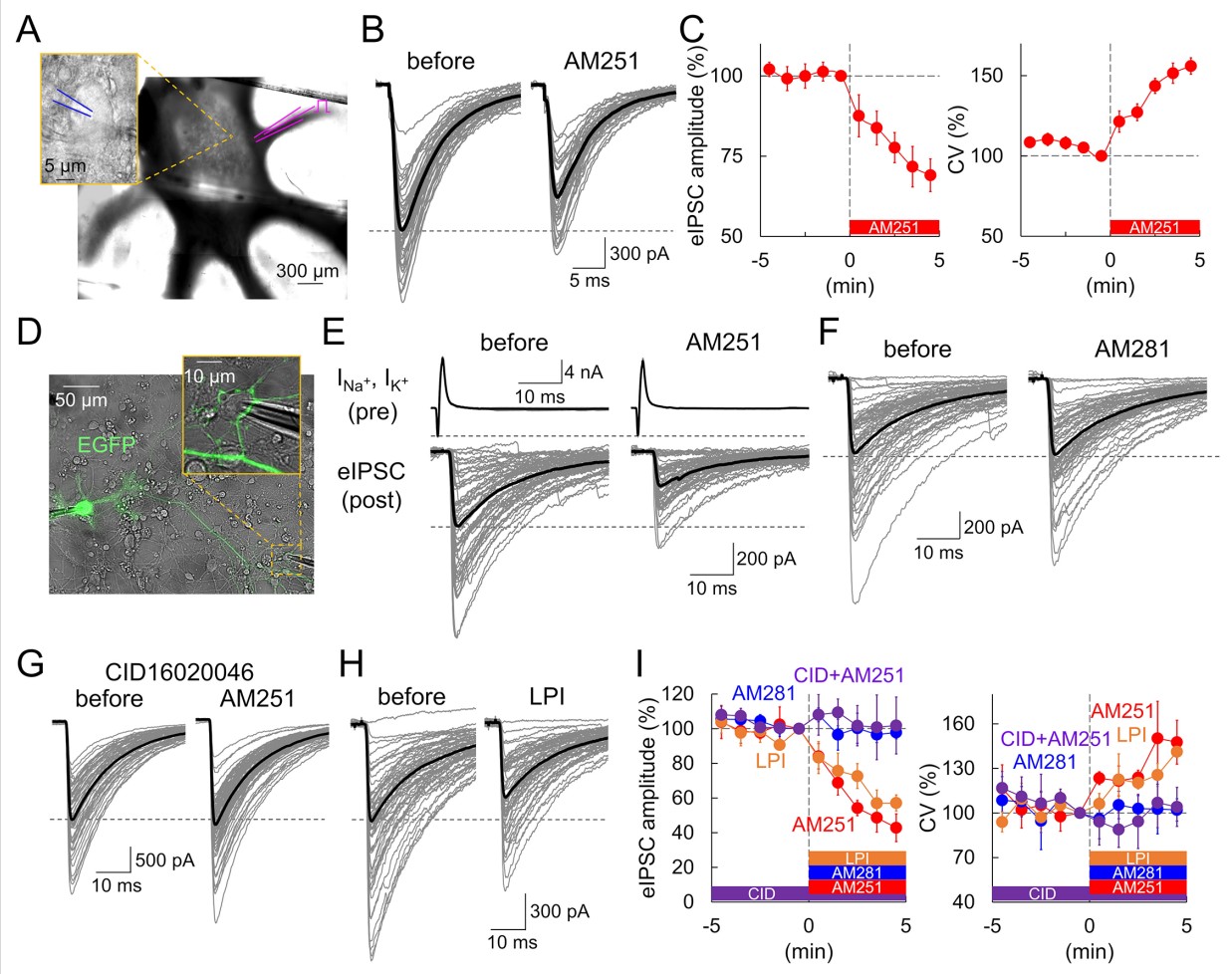

**Figure 1.** Reduced transmission at PC-DCN synapses by GPR55 in slice and culture. (**A**) Image of patch-clamp recording from a DCN neuron in slice. A magnified image of the recorded neuron is shown as an inset (a patch pipette is indicated in blue). A pipette for electrical stimulation (magenta) was placed at the white matter. (**B, C**) Individual (gray) and averaged (black) traces (**B**), and time courses of eIPSC amplitude (**C**, left) and CV (**C**, right) before and after the AM251 application (at time 0), n=6. (**D**) Image of dual whole-cell recordings from a presynaptic EGFP-labeled PC soma and its postsynaptic target neuron in culture. (**E–H**) Representative traces of eIPSCs before and 5 min after the application of AM251 (**E**), AM281 (**F**), AM251 in the presence of CID (**G**), or LPI (**H**). In (**E**), presynaptic Na$^+$ and the following K$^+$ currents (INa$^+$, IK$^+$) at a presynaptic PC soma are also shown. (**I**) Time courses of eIPSC amplitude and CV. n=6 (AM251), 5 (AM281), 5 (CID + AM251), and 5 cells (LPI). CID was applied from the beginning of the recording. The internal solution contained 0.5 mM EGTA. Data are shown as mean ± SEM.

The online version of this article includes the following source data and figure supplement(s) for figure 1:

**Source data 1.** Data for eIPSCs and Na$^+$ currents shown in *Figure 1* and *Figure 1—figure supplement 1*.

**Figure supplement 1.** No effects of GPR55 on IPSCs time courses and somatic Na $^+$ currents.

---

*Wilson and Nicoll, 2001*; *Diana et al., 2002*). Aiming at much more detailed analysis of synaptic function based on the feasibility to perform direct patch-clamp recordings from axon terminals, we also tested the action of GPR55 on PC outputs using the primary culture preparation. To fluorescently visualize PCs, we used an adeno-associated virus (AAV) vector, AAV2-CA-EGFP that preferentially infects PCs among cerebellar neurons, as in previous studies (*Kawaguchi and Sakaba, 2015*). Simultaneous whole-cell somatic patch-clamp recordings were performed from a presynaptic PC and its postsynaptic neuron (possible DCN neuron) surrounded by lots of EGFP-positive PC boutons, in the presence of NBQX (10 µM) (*Figure 1D*). Presynaptic PCs were voltage-clamped at –70 mV, and Na$^+$ current escaping as an AP from the patched soma to the axon was elicited by applying a voltage pulse (to 0 mV, 1–5 ms), which evoked IPSCs in the voltage-clamped postsynaptic target cell. In line with the data obtained in slices, synaptic transmission decreased upon GPR55 activation with AM251 (500 nM), manifested as a decrease in IPSC amplitudes (42.9 ± 8.0% at +5 min, p=0.0134) and an increase in the

CV (147.6 ± 14.7% at +5 min, p=0.0431) (*Figure 1E and I*). The effect of AM251 became evident a few minutes after application and somehow reached a saturated level within 5 min in both slice and culture preparations. It should be noted that AM251 is known to act not only as a GPR55 agonist, but also as an inverse agonist for CB1 (*Ryberg et al., 2007*; *Henstridge et al., 2008*). To confirm that the effect of AM251 is not due to the CB1 receptor inhibition, we also tested AM281, a CB1-specific inverse agonist. Application of AM281 (500 nM) did not induce any significant changes in eIPSC amplitude or its CV (amplitude: 97.8 ± 3.6% at +5 min, p=0.737; CV: 102.1 ± 7.5% at +5 min, p=0.260) (*Figure 1F and I*), suggesting that the synaptic depression caused by AM251 is not due to CB1 receptor inhibition. Considering a previous study showing that PC synapses exhibit no modulation by CB1 receptors (*Hirono and Yanagawa, 2021*; see also *Figure 1F and I*), the effects of AM251 on synaptic outputs from a PC bouton seemed to be ascribed to activation of GPR55. Indeed, the suppressive effect of AM251 on synaptic transmission was abolished by a GPR55 antagonist CID16020046 (CID, 5 μM; $IC_{50}$=0.21 μM, specific for GPR55 without affecting CB1 receptor) (amplitude: 101.9 ± 16.4% at +5 min, p=0.949; CV: 104.1 ± 13.1% at +5 min, p=0.624) (*Figure 1G and I*). In addition, we also tested another GPR55 ligand, lysophosphatidylinositol (LPI, 1 μM; $EC_{50}$=30 nM, *Oka et al., 2007*), which is known as an endogenous ligand of GPR55 (*Oka et al., 2007*; *Henstridge et al., 2008*). Several types of LPI are known, and here we used soy-derived LPI (Merck, containing approximately 58% C16:0 and 42% C18:0 or C18:2), which was shown to activate GPR55 for synaptic modulation in the hippocampus (*Rosenberg et al., 2023*). As shown in *Figure 1H and I*, similar reduction of eIPSC amplitude and increase in its CV to those by AM251 were observed after the extracellular application of LPI (amplitude: 57.2 ± 4.6% at +5 min, p=0.00111; CV: 141.2 ± 3.6% at +5 min, p=0.00292). Neither application of AM251 nor LPI changed the time courses of IPSCs (AM251, 20–80% risetime: 0.81±0.05 to 0.77±0.04 ms, p=0.296; LPI, 0.71±0.10 to 0.65±0.12 ms, p=0.222) (*Figure 1—figure supplement 1A*) and presynaptic $Na^+$ currents recorded at the PC soma (AM251, 3.30±0.33 to 3.25±0.40 nA, p=0.520; LPI, 2.87±0.30 to 2.89±0.31 nA, p=0.728) (*Figure 1—figure supplement 1B and C*). Taken all these results based on pharmacological agents for GPR55 together, it was suggested that synaptic transmission is negatively regulated by GPR55 at synapses formed by PC axon terminals, presumably acting on the presynaptic side.

It is a critical issue to examine the presence of GPR55 at PC axon terminals. To address this, we attempted to fluorescently label neurons in cerebellar culture with molecules targeting GPR55, using a fluorescent derivative of AM251, Tocrifluor T1117 (T1117; *Daly et al., 2010*; *Sylantyev et al., 2013*). As shown in *Figure 2A and B*, confocal fluorescent images of GFP-positive PC axons and terminals clearly showed significant T1117 signal after incubation with T1117 (20 nM) for 2 min, which was selectively suppressed by the CRISPR/Cas9-mediated acute knock-down of GPR55 expression (relative fluorescence density of T1117, mean ± SD: control, 1.00±0.49; GPR55-KD, 0.53±0.27, p<0.001; Cas9-alone, 1.04±0.47, p=0.493, compared with control, Dunnett's test) (*Figure 2C*; see 'Materials and methods' for details of the CRISPR/Cas9 knock-down). Thus, T1117 signals at PC axon terminals would be ascribed at least partially to GPR55. It should be noted that PC boutons showed marked deviation in the T1117 fluorescence intensity, probably reflecting variable GPR55 abundance at individual PC boutons. In addition, T1117 signals were also evident at GFP-negative structures (see *Figure 2*), probably due to labeling of axons/terminals of neurons such as granule cells and inhibitory interneurons which are well-known to potently express CB1 (*Tsou et al., 1998*), which is antagonized by AM251. Indeed, GFP-negative structures neighboring the GPR55-knocked down GFP-positive PC boutons showed clear T1117 signals (*Figure 2B and C*). The above results showing PC terminals labeled with a fluorescent AM251 analogue depending on the GPR55 gene expression indicated the presence of GPR55 with variable density at individual PC boutons, supporting the idea about modulation of PC output synapses by GPR55 (see *Figure 1*).

## GPR55 suppresses transmitter release in PC boutons without changing presynaptic $Ca^{2+}$ influx

Taking advantage of the feasibility to directly patch-clamp record from GPR55-expressing PC axon terminals (*Kawaguchi and Sakaba, 2015*), we next attempted to examine the mechanism how GPR55 suppresses the transmission. One candidate mechanism is the reduction of the presynaptic $Ca^{2+}$ influx, as CB1 does at the calyx of Held synapses and climbing fiber synapses on PCs (*Kreitzer and Regehr, 2001*; *Kushmerick et al., 2004*; *Wu et al., 2020*). To record presynaptic $Ca^{2+}$ current, direct

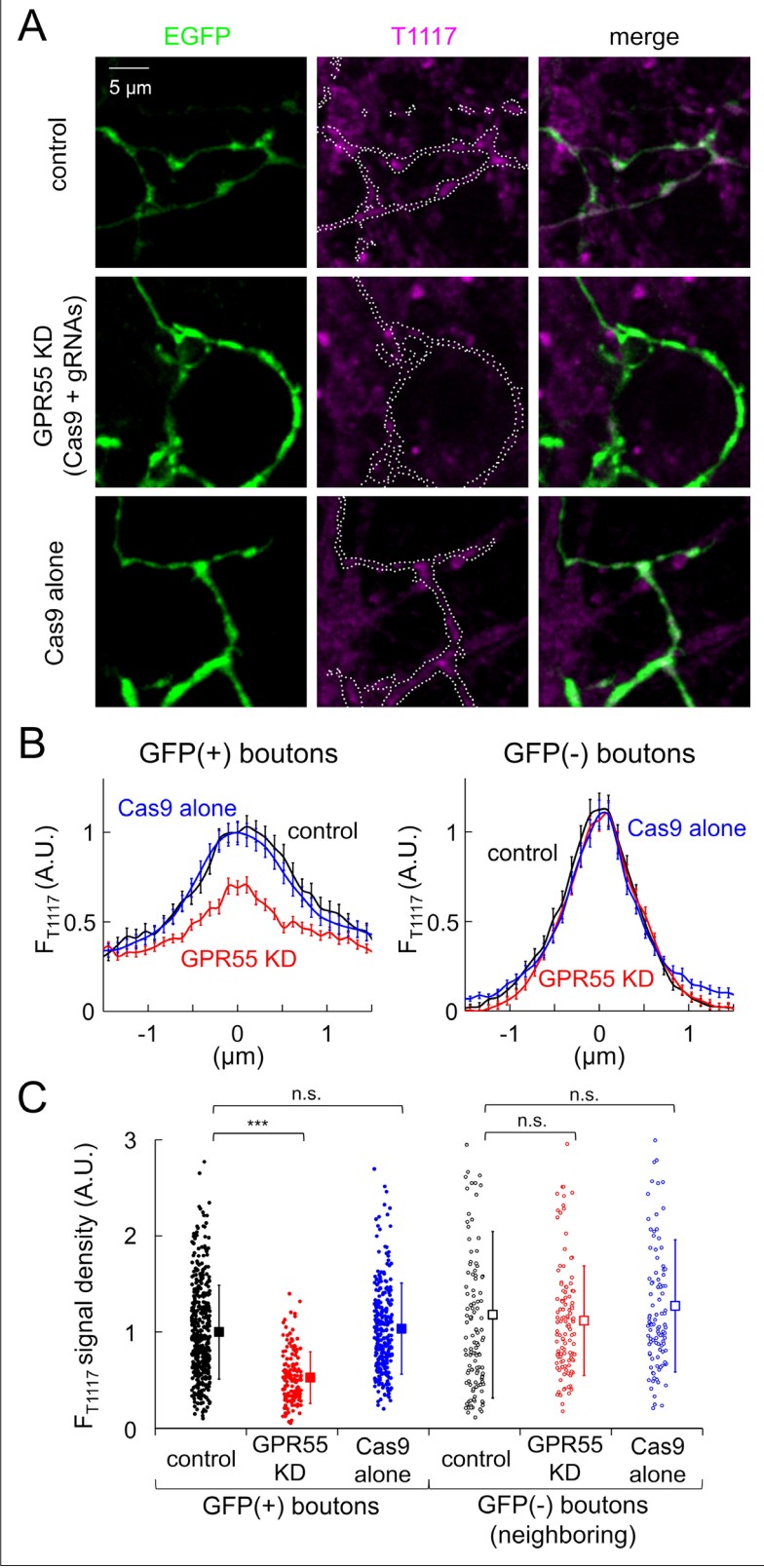

**Figure 2.** Localization of GPR55 at PC axonal boutons. (**A**) Representative confocal fluorescent images of GFP and T1117 labeling in control and transfected cells with GPR55-targeting CRISPR/Cas9 vectors or Cas9-alone. White dotted lines in the T1117 images indicate the outlines of GFP-labeled axons. (**B**) Spatial profiles of T1117 signals in GFP(+) or GFP(-) boutons. Averaged data are shown as mean ± SEM. Fluorescence peak of GFP (for

*Figure 2 continued on next page*

*Figure 2 continued*

GFP(+) boutons) or T1117 (for GFP(-) boutons) was aligned to the position 0. (**C**) $F_{T1117}$ signal density (averaged fluorescence per pixel) in GFP(+) boutons (control, 455; GPR55-KD, 161; Cas9-alone, 270 boutons) or that in GFP(-) boutons near GFP(+) ones (control, 124; GPR55-KD, 116; Cas9-alone, 104 boutons). Data for individual boutons (circles) and mean ± SD (squares) are shown. ***p<0.001; n.s., not significant.

The online version of this article includes the following source data for figure 2:

**Source data 1.** T1117 fluorescence data for **Figure 2**.

voltage-clamp recordings were performed from PC boutons (**Figure 3A**). Concomitantly, we also monitored the membrane capacitance ($C_m$) increase ($\Delta C_m$) caused by the $Ca^{2+}$ influx, which is an indication of the amount of exocytosed synaptic vesicles. Unexpectedly, $Ca^{2+}$ currents induced by square depolarizing pulses (0 mV for 5 ms) were not affected by AM251 in the voltage-clamped PC boutons (53.0±5.6 to 52.4±6.3 pA/pF, p=0.808), whereas the resultant increase in $C_m$ was markedly diminished (22±4 to 13±2 fF/pF, p=0.0448) (**Figure 3B and C**). This strong reduction of vesicle exocytosis by AM251 was mediated by GPR55, as evidenced by its abolishment by the knock-down of GPR55 expression using CRISPR/Cas9 system without any effects on presynaptic $Ca^{2+}$ influx (**Figure 3D and E**; $Ca^{2+}$ current: Cas9 alone, 55.1±9.8 to 54.7±9.6 pA/pF, p=0.712, paired $t$-test; GPR55-KD, 56.7±10.0 to 54.7±9.8 pA/pF, p=0.175; $\Delta C_m$: Cas9 alone, 28±1 to 15±1 fF/pF, p<0.001; GPR55-KD, 27±5 to 24±6 fF/pF, p=0.185). Taken together, our direct patch-clamp recordings from PC terminals indicated potent suppression of vesicle exocytosis by GPR55 without changing $Ca^{2+}$ influx, which is distinct from the typical synaptic modulation by CB1 receptors through a reduction of presynaptic $Ca^{2+}$ influx.

We further tested whether GPR55 also affects APs and postsynaptic $GABA_A$ receptor responsiveness at PC-DCN synapses. Previous studies demonstrated that AP waveforms are subject to

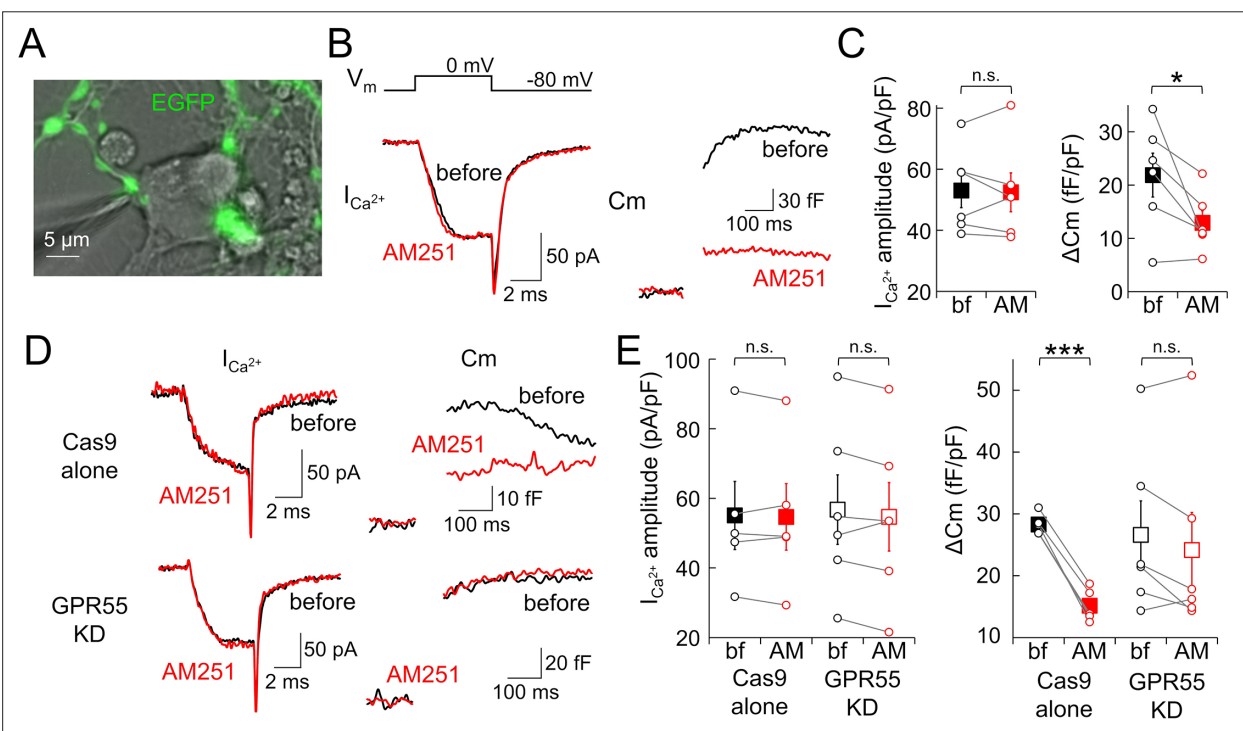

**Figure 3.** Reduced vesicle exocytosis by GPR55 without changing presynaptic $Ca^{2+}$ influx. (**A**) Image of direct patch-clamp recording from an EGFP-labeled PC axon terminal in culture. (**B–E**) Representative traces (**B, D**) and amplitude of presynaptic $Ca^{2+}$ currents during 5 ms depolarization ($ICa^{2+}$) and the resultant $C_m$ increase ($\Delta C_m$, **C, E**) before and after the AM251 application in wild-type (n=6) (**B, C**) or transfected cells with Cas9-alone (n=5), or CRISPR/Cas9 for GPR55 (n=6) (**D, E**). Both $ICa^{2+}$ and $\Delta C_m$ were normalized by the size of presynaptic $C_m$ under the voltage-clamp. The internal solution contained 0.5 mM EGTA. Data are mean ± SEM. *p<0.05; ***p<0.001; n.s., not significant.

The online version of this article includes the following source data for figure 3:

**Source data 1.** Data for presynaptic $ICa^{2+}$ and $\Delta C_m$ shown in **Figure 3**.

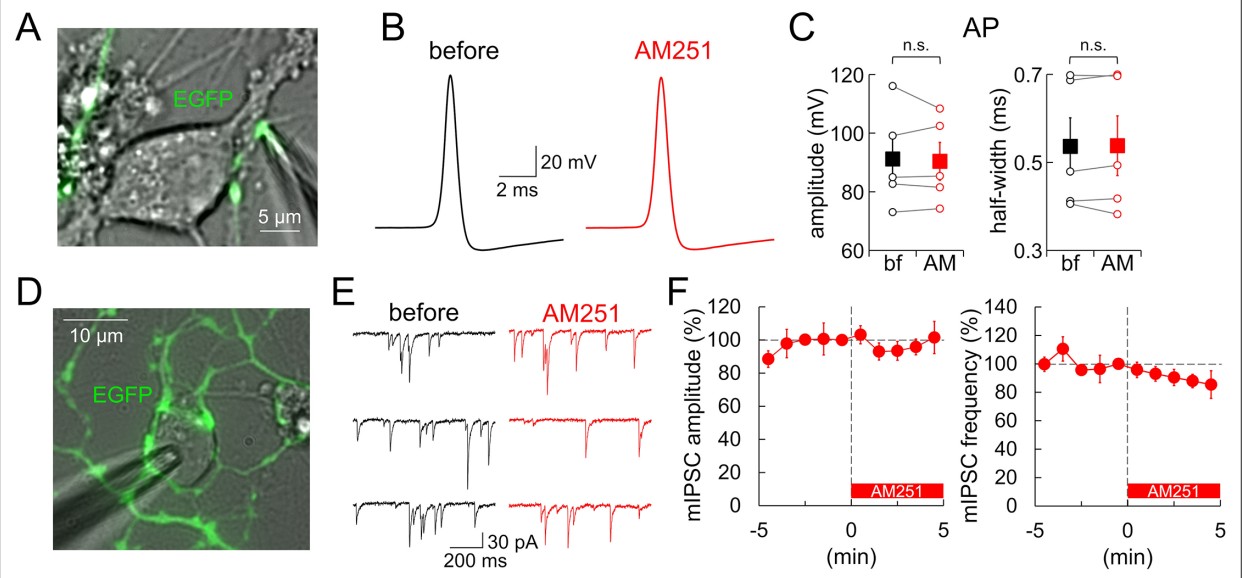

**Figure 4.** GPR55 has no effect on APs or postsynaptic responsiveness. (**A**) Image of direct recording from an EGFP-labeled PC axon terminal in culture. (**B, C**) Representative traces (**B**), amplitude and half-width (**C**) of APs recorded from PC terminals before and 5 min after the AM251 application (n=5). (**D**) Image of mIPSC recordings from a neuron innervated by lots of EGFP-positive PC boutons in culture. (**E**) Representative traces of mIPSCs before and 5 min after the AM251 application. (**F**) Time courses of mIPSC amplitude and frequency (n=4). At 0 min, AM251 was applied. The internal solution contained 0.5 mM EGTA. Data are shown as mean ± SEM. n.s., not significant.

The online version of this article includes the following source data and figure supplement(s) for figure 4:

**Source data 1.** Data for APs and mIPSCs shown in *Figure 4* and *Figure 4—figure supplement 1*.

**Figure supplement 1.** No effects of GPR55 on time courses of mIPSCs 20–80% risetime or half-width of mIPSCs before and after application of AM251 (n=5).

modulation in PC axons in a manner dependent on high frequency firing, local activation of axonal transmitter receptors, and/or intracellular cAMP changes (*Kawaguchi and Sakaba, 2015*; *Zorrilla de San Martin et al., 2017*; *Furukawa et al., 2025*). To test the possibility of a change in the AP waveform by GPR55, we performed direct patch-clamp recordings of APs from a presynaptic bouton of a PC axon (*Figure 4A*). As shown in *Figure 4B and C*, current-clamp recordings from the axon terminals detected no changes of APs by GPR55 in both the amplitude and time course (amplitude: 91.2±7.5 to 90.4±6.4 mV, p=0.701, paired *t*-test; half-width: 0.54±0.06 to 0.54±0.07 ms, p=0.832). In terms of modulation of postsynaptic responsiveness by GPR55, a decrease in accumulation of postsynaptic $GABA_A$ receptors at hippocampal GABAergic synapses was suggested (*Khan et al., 2018*; *Rosenberg et al., 2023*). However, miniature IPSCs (mIPSCs) recorded from a postsynaptic DCN cell innervated by lots of EGFP-labeled PC boutons in the presence of TTX (1 µM) and NBQX (10 µM) (*Figure 4D*, *Figure 4—figure supplement 1*) showed no change in either amplitude or frequency after the AM251 application (amplitude: 101.5 ± 9.7% at +5 min, p=0.941; frequency: 85.4 ± 13.4%, p=0.608) (*Figure 4E and F*). Together, these results demonstrated that GPR55 suppresses transmitter release from PC boutons not by altering presynaptic $Ca^{2+}$ influx, AP waveforms, or postsynaptic responsiveness, but rather by directly reducing vesicle exocytosis.

## GPR55 decreases the amount of vesicles in the readily releasable pool

To obtain insights into the mechanism of GPR55-mediated suppression of neurotransmitter release, we biophysically analyzed the relation between presynaptic $Ca^{2+}$ influx and vesicle exocytosis by systematically changing depolarization pulses to a PC bouton. Depolarizing pulses with different lengths altered the duration of presynaptic $Ca^{2+}$ current, leading to distinct amounts of vesicular release (*Figure 5A*). As summarized in *Figure 5B*, the $Ca^{2+}$ current amplitude and its activation kinetics were again not significantly affected by GPR55 activation (amplitude, p=0.402; tau, p=0.339, ANOVA), supporting the idea that GPR55 does not affect the presynaptic $Ca^{2+}$ influx. On the other hand, $\Delta C_m$ induced by any duration of depolarization pulses showed ~50% reduction in the presence of AM251 (31±6 fF/pF

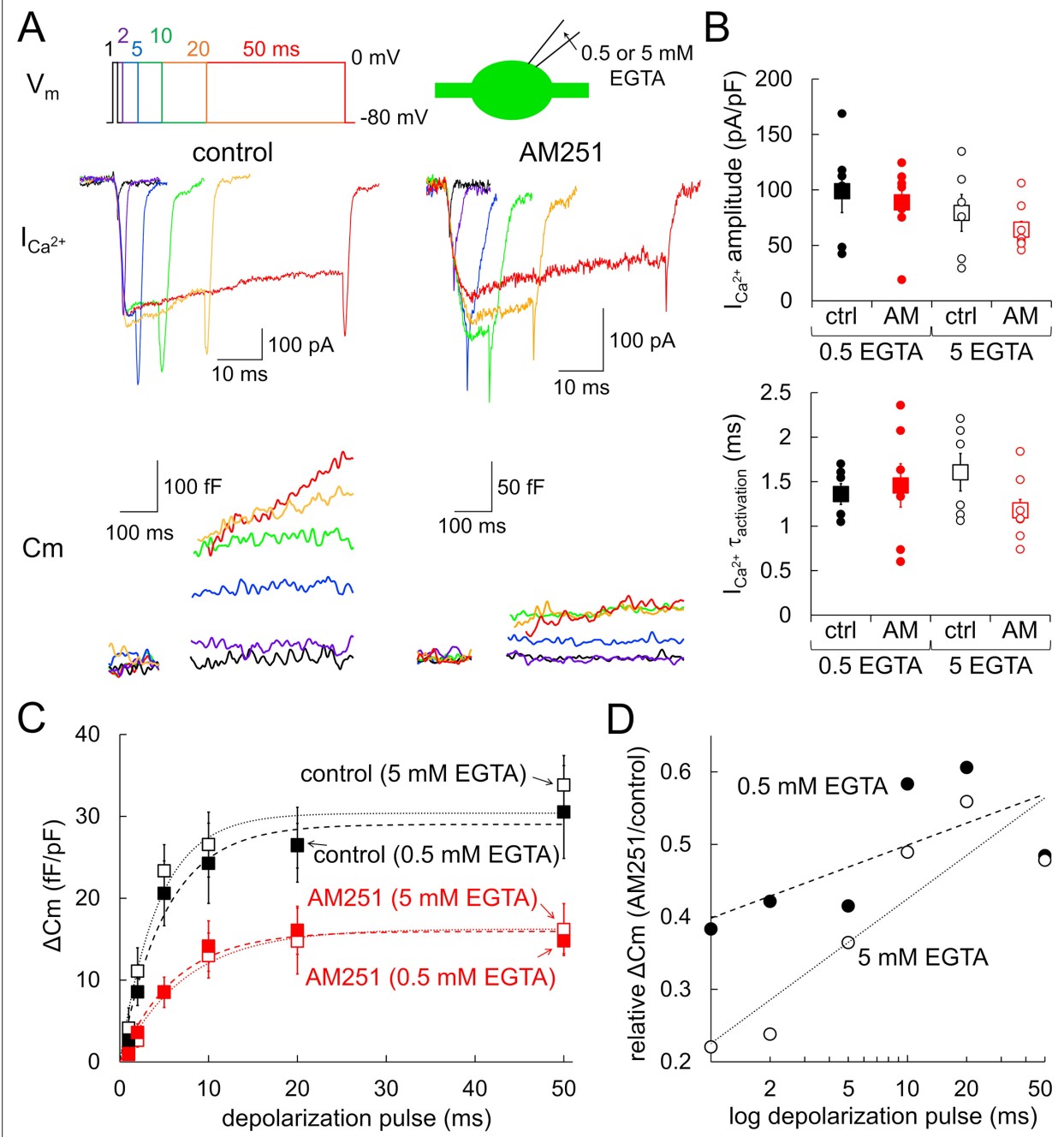

**Figure 5.** Decrease in RRP vesicles by GPR55. (**A**) Representative traces of presynaptic $ICa^{2+}$ (middle) and the resultant $C_m$ increase (bottom) recorded with presynaptic 0.5 mM EGTA upon 1, 2, 5, 10, 20, or 50 ms of depolarization pulses (to 0 mV) (top) without (left) or with (right) AM251. Experiments were performed with a presynaptic patch pipette containing 0.5 mM or 5 mM EGTA. (**B, C**) Amplitude and time constant for activation of $ICa^{2+}$ (**B**), and $C_m$ change (**C**) upon depolarization pulses recorded without (black, 0.5 mM EGTA, n=6; 5 mM EGTA, n=6) or with AM251 (red, 0.5 mM EGTA, n=7; 5 mM EGTA, n=8). Single exponential fits for each are shown as dotted lines. Both $ICa^{2+}$ and $C_m$ change were normalized by the size of presynaptic $C_m$ under the voltage-clamp. In (**B**), data for individual cells (circles) and mean ± SEM (squares) are shown. (**D**) Ratio of $C_m$ increase ($C_m$ increase with AM251 divided by that in control) recorded with 0.5 or 5 mM presynaptic EGTA is plotted against the logarithm of pulse duration.

The online version of this article includes the following source data and figure supplement(s) for figure 5:

**Source data 1.** Data for $ICa^{2+}$ and $\Delta C_m$ shown in *Figure 5* and *Figure 5—figure supplement 1*.

**Figure supplement 1.** AM251 and LPI occlude the suppressive effects on vesicle release each other.

**Figure supplement 2.** GPR55-mediated suppression of release in PC boutons with 5 mM EGTA.

and 15±2 fF/pF upon 50 ms pulse without and with AM251, respectively, p=0.0289, Tukey–Kramer test) (*Figure 5C and D*). Thus, GPR55 halves the number of synaptic vesicles released within tens of milliseconds after a strong $Ca^{2+}$ influx, corresponding to the readily-releasable pool (RRP), presumably consisting of immediately releasable and slowly releasable vesicles as in other boutons such as calyx of Held (*Sakaba, 2006*). The decrease of RRP vesicles could be the primary cause for the decrease in AP-triggered synaptic transmission. We also examined whether LPI, an endogenous agonist for GPR55, exerts a similar effect. When GPR55 was activated by LPI, $\Delta C_m$ was smaller in size (17±2 fF/pF upon 50 ms pulse, p=0.0488, compared with control, Tukey–Kramer test) to an extent of that by AM251, which was not enhanced by additional application of AM251 (16±4 fF/pF upon 50 ms pulse, p=1.00, compared with LPI alone, Tukey–Kramer test) (*Figure 5—figure supplement 1*). These results indicate that both LPI and AM251 act through GPR55 and reduce the RRP.

Taking into consideration that the cytoplasm of PCs contains extremely potent $Ca^{2+}$ buffering capacity due to the large amount of calbindin (*Fierro and Llano, 1996*; *Bornschein et al., 2013*), releasable vesicles in PC boutons might show steep dependency of their release competency on their position relative to the voltage-gated $Ca^{2+}$ channels (VGCCs). Previous studies reported that vesicular fusion is insensitive to exogenous EGTA in the intracellular solution, a relatively slow but high-affinity $Ca^{2+}$ buffer, suggesting that vesicles tightly coupled to VGCCs are release-competent in PC boutons (*Kawaguchi and Sakaba, 2015*; *Díaz-Rojas et al., 2015*). Indeed, the RRP size was comparable between 0.5 and 5 mM cytoplasmic EGTA conditions (0.5 mM EGTA, 31±6 fF/pF; 5 mM EGTA, 34±4 fF/pF upon 50 ms pulse, p=0.928, Tukey–Kramer test) (*Figure 5C*, *Figure 5—figure supplement 2*). On the other hand, previous studies showed a decrease in release by EGTA at synapses with relatively loose coupling between VGCCs and releasable vesicles, such as squid giant synapses, calyx of Held synapses, or hippocampal mossy fiber synapses (*Adler et al., 1991*; *Borst and Sakmann, 1996*; *Vyleta and Jonas, 2014*). Therefore, if GPR55 loosens the $Ca^{2+}$-release coupling, EGTA might become influential to the release even in PC boutons. However, the increase in cytoplasmic EGTA from 0.5 to 5 mM did not reduce the apparent RRP size in PC boutons when GPR55 was activated by AM251 (5 mM EGTA, 16±3 fF/pF upon 50 ms pulse with AM251, p=0.992, compared to 15±2 fF/pF with 0.5 mM EGTA and AM251, Tukey–Kramer test) (*Figure 5C*). Notably, plotting the fold reduction

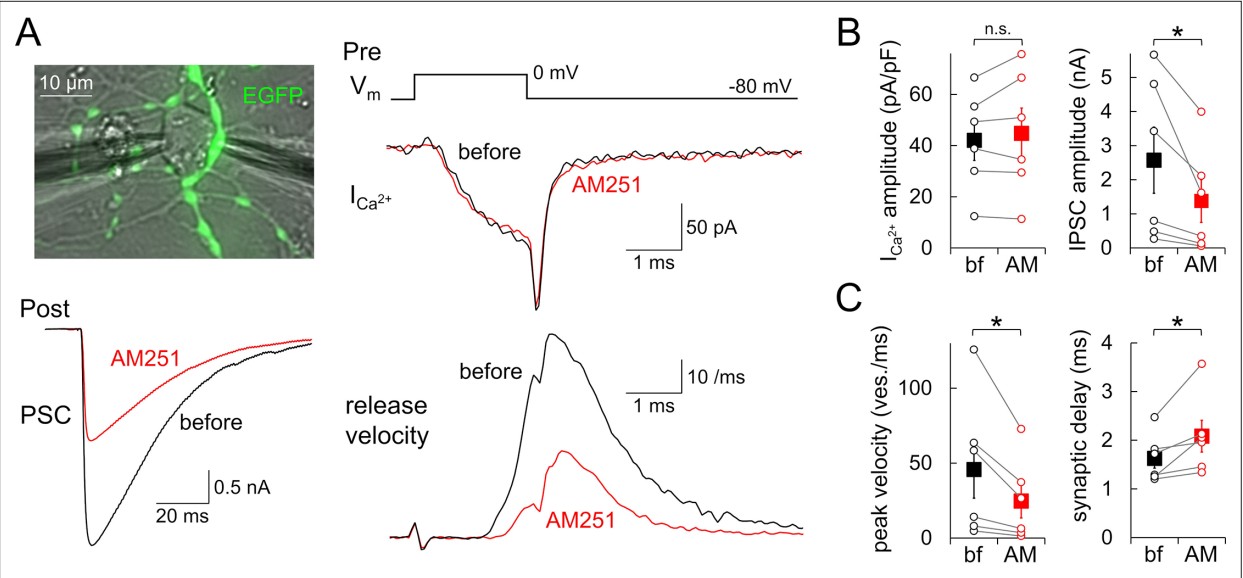

**Figure 6.** Halved velocity of exocytosis shown by pre- and postsynaptic paired recordings. (**A**) Image of paired recordings from a presynaptic EGFP-labeled PC bouton and its postsynaptic neuron (upper left), and representative traces of the presynaptic $I_{Ca^{2+}}$ (upper right), IPSC (lower left), and velocity of vesicular fusion (bottom right; calculated by the deconvolution of IPSC trace) upon 2ms depolarization before and after the AM251 application. (**B, C**) Amplitude of $I_{Ca^{2+}}$ (**B**, left, calibrated by the presynaptic $C_m$ under the voltage-clamp), IPSC (**B**, right), maximal release velocity (**C**, left), and synaptic delay (**C**, right) before and after application of AM251. The internal solution contained 0.5 mM EGTA. Data are mean ± SEM. *p<0.05; n.s., not significant. n=6 pairs.

The online version of this article includes the following source data for figure 6:

**Source data 1.** Data for $I_{Ca^{2+}}$, IPSCs, and release velocity shown in *Figure 6*.

of $\Delta C_m$ by AM251 against the duration of depolarization suggested that a higher amount of EGTA augmented the suppressive effect of GPR55, particularly for the release in response to short pulses (*Figure 5D*). This result implies that the reduction of release by GPR55 might be somehow related to the $Ca^{2+}$-release coupling, although its extent is too small to be reflected as a clear change of release sensitivity to EGTA. In summary, based on biophysical analysis of $Ca^{2+}$-dependent release in terms of the modulations by GPR55 and cytoplasmic EGTA, we conclude that the GPR55 activation reduces the number of vesicles in the RRP, which might be accompanied by a slight change of the functional coupling between release and $Ca^{2+}$ influx.

## Slowed vesicular fusion by GPR55

To further investigate the GPR55-mediated downregulation of vesicular fusion, the $Ca^{2+}$-activated release kinetics were biophysically analyzed by simultaneous patch-clamp recordings from both a presynaptic PC bouton and a postsynaptic cell. To avoid saturation of postsynaptic $GABA_A$ receptors, we applied a short depolarization pulse (0 mV, 2 ms) to the patched bouton and recorded presynaptic $Ca^{2+}$ and postsynaptic currents (*Figure 6A*). In line with the above results, presynaptic $Ca^{2+}$ current showed little change by the AM251 administration (42.1±7.9 to 44.8±9.9 pA/pF, p=0.332, paired *t*-test), while postsynaptic currents were clearly reduced by half (2.57±0.97 to 1.38±0.63 nA, p=0.0448, paired *t*-test) (*Figure 6A and B*). By deconvolving the postsynaptic currents with the template mIPSC trace, we estimated the temporal change of vesicular fusion velocity (for detail, see 'Materials and methods'; *Kawaguchi and Sakaba, 2015*). The peak of vesicular fusion velocity was almost halved by AM251 without a marked change of time course (45.7±19.1 to 24.7±11.3 vesicles/ms, p=0.0467, paired *t*-test) (*Figure 6C*), showing similar reduction to that observed in the apparent RRP size (vesicular fusion velocity: 46.9 ± 4.4%; $\Delta C_m$: 48.4% upon 50 ms pulse shown in *Figure 5D*). Thus, it would be reasonable to conclude that the RRP reduction, but not a change in the $Ca^{2+}$-activated release probability, predominantly underlies the synaptic suppression induced by GPR55. However, notably, AM251 induced a modest yet significant increase in synaptic delay, estimated by the time to the onset of release (onset of vesicular release: 1.63±0.20 to 2.08±0.33 ms, p=0.0403, paired *t*-test) (*Figure 6C*). Thus, together with the idea that GPR55 somehow lowers the effectiveness of $Ca^{2+}$-triggered release (as shown in *Figure 5D*), the apparent RRP reduction by GPR55 in PC boutons might be caused by a process accompanied with a slight downregulation of the $Ca^{2+}$-mediated activation process of vesicular fusion.

## Reduction of AP-sensitive vesicles by GPR55

The above experiments demonstrated that GPR55 suppresses synaptic transmission primarily by reducing the RRP size without altering $Ca^{2+}$ influx at the presynaptic terminal. To discriminate two possible causes for the decrease in RRP size: relative increase in the fraction of reluctant vesicles or reduction of total vesicles present at the presynaptic terminal, we performed fluorescence imaging of vesicular fusion with synapto-pHluorin expressed in PCs using AAV or microinjection of plasmid DNA into the nucleus. First, APs were repetitively evoked (400 APs at 20 Hz) at the voltage-clamped PC soma by depolarization pulses triggering somatic $Na^+$ currents (*Figure 7—figure supplement 1*), and the fluorescence changes of pHluorin at axon terminals were monitored. Synapto-pHluorin shows little fluorescence upon light illumination when its pH-sensitive fluorophore is exposed to the acidic vesicular lumen and becomes fluorescent after exocytosis through exposure to the external neutral pH environment (*Figure 7A*), providing the sign of vesicle fusion as a fluorescence increase (*Sankaranarayanan and Ryan, 2000*). As shown previously (*Kawaguchi and Sakaba, 2015*), PC axonal varicosities exhibited synapto-pHluorin fluorescence increase by 23.3 ± 2.0% upon 400 APs at 20 Hz (*Figure 7B–D*). After the increase in fluorescence during the AP train stimulation, pHluorin signal decayed with a time constant of about 40 s, reflecting the endocytosis of exocytosed synapto-pHluorin and the subsequent re-acidification of the vesicular lumen (*Nicholson-Tomishima and Ryan, 2004*; *Yamashita et al., 2018*). In accord with the GPR55-mediated reduction of synaptic transmission (see *Figure 1*), AM251 administration reduced the pHluorin signal increase evoked by the 400 APs ($\Delta F_{400APs}$) by 53.8 ± 17.0% on average (p<0.001, by paired *t*-test), with no change in its decay time course ($\tau_{decay}$: 42.1±7.8–40.9±4.5 s, p=0.780, paired *t*-test) (*Figure 7C*). The AP-triggered increase in pHluorin fluorescence was specific to axonal varicosities, and the GPR55 activation reduced the size of the fluorescence increase with substantial variability among individual boutons (*Figure 7B–D*). Altered

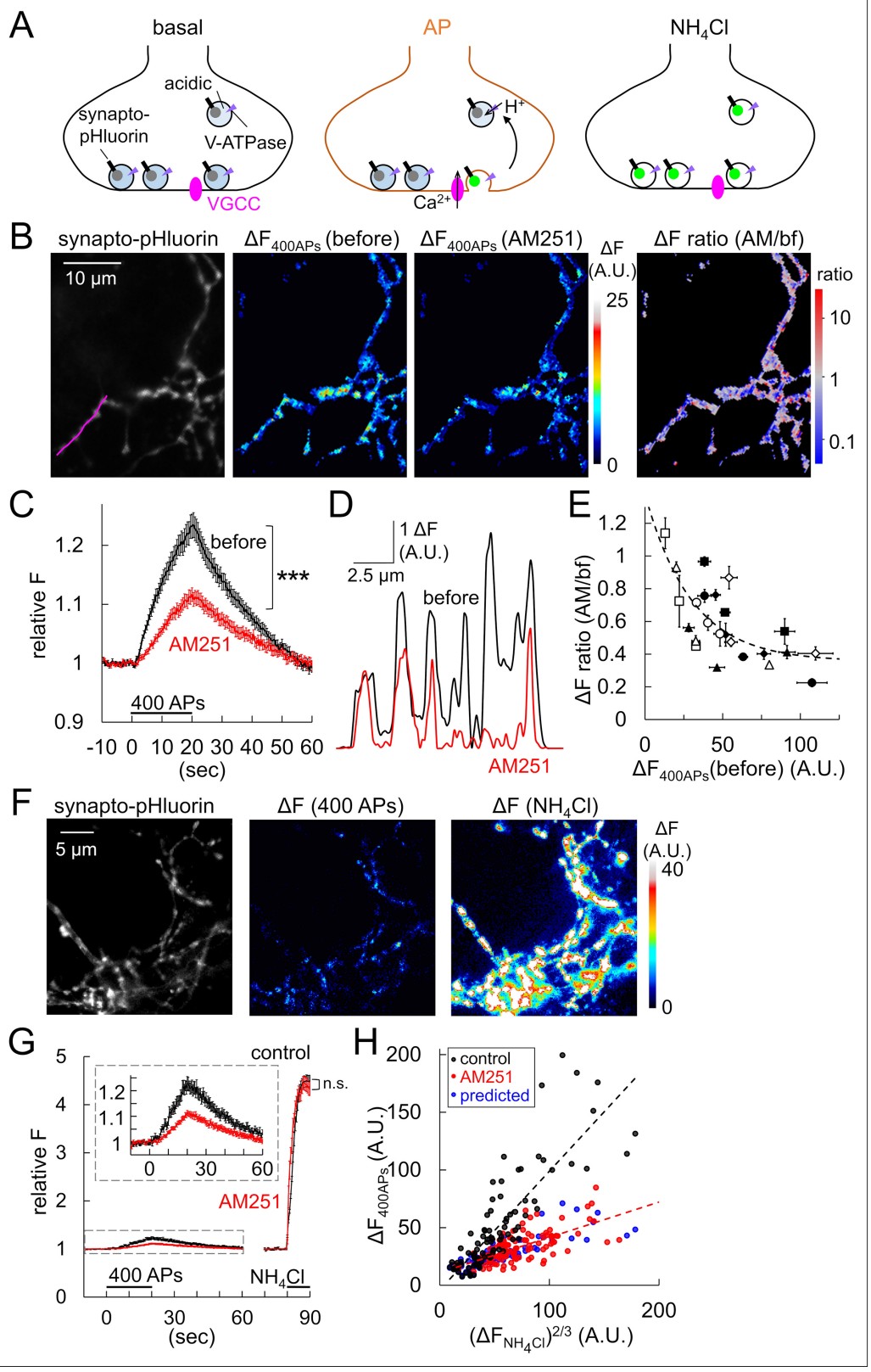

**Figure 7.** Halved vesicular releases by GPR55 imaged with pHluorin. (**A**) Schematic illustration for imaging of vesicle exocytosis with synapto-pHluorin. Upon vesicular fusion at the AP arrival, pHluorin becomes fluorescent because of exposure to the neutral pH. $NH_4^+$ application forcefully neutralizes vesicle lumen. (**B**) Representative images of synapto-pHluorin fluorescence and the color-coded fluorescence increase upon 400 APs ($\Delta F_{400APs}$)

*Figure 7 continued on next page*

*Figure 7 continued*

in PC terminals before and after the AM251 application. The right-most panel shows the ratio of fluorescence increase after the AM251 application relative to that before, represented in pseudo-color. (**C**) Time courses of 400 APs-triggered pHluorin fluorescence changes before and after the AM251 application. n = 54 boutons. (**D**) Plot of $\Delta F_{400APs}$ along a PC axon (indicated as a magenta line in **B**) before and after the AM251 application. (**E**) Ratio of $\Delta F_{400APs}$ after the AM251 application relative to that before is plotted against the $\Delta F_{400APs}$ before the AM251 application. Data from different cells are shown as different symbols. n=8 cells. Dashed line represents data fitting with a function: $y=1.02e^{-0.0334x} + 0.355$, $R^2=0.506$. (**F**) Images of pHluorin fluorescence and the color-coded fluorescence increases in a PC axon upon 400 APs and the following $NH_4Cl$ application (50 mM). (**G**) Time courses of synapto-pHluorin fluorescence changes upon 400 APs and the following $NH_4Cl$ application without or with AM251 (control, n=47 boutons; AM251, n=70). Inset shows enlarged traces. (**H**) $\Delta F_{400APs}$ is plotted against the 2/3rd of $\Delta F$ caused by $NH_4Cl$ ($\Delta F_{NH4Cl}$) for individual boutons without or with AM251. The effect of AM251 was predicted (blue) by conversion of the dataset for control (black) based on the relationship shown in (**E**), showing similar distribution to the actual data obtained with AM251 (red). Fitted line for control: $y=1.03 x - 4.05$, $R^2=0.704$; for AM251, $y=0.307 x+10.9$, $R^2=0.441$. Data are mean ± SEM. \*\*\*p<0.001; n.s., not significant.

The online version of this article includes the following source data and figure supplement(s) for figure 7:

**Source data 1.** Data for pHluorin fluorescence and $Na^+$ currents shown in *Figure 7* and *Figure 7—figure supplement 1*.

**Figure supplement 1.** No effects of GPR55 on somatic $Na^+$ currents upon repetitive stimulation at 20 Hz.

abundance of GPR55 in distinct boutons might be responsible for the variable effect of AM251 (see *Figure 2*). Interestingly, however, plotting the extent of release reduction by AM251 against the original size of $\Delta F_{400APs}$ indicated that boutons showing a larger amount of release tend to be more susceptible to suppression by the GPR55 activation (*Figure 7E*). This indicates that the variable control of the synaptic output exerted by GPR55 depends on the original amount of release in individual boutons along a PC axon. On the other hand, when the intracellular vesicles with an acidic internal lumen were ubiquitously neutralized by exogenous application of $NH_4Cl$ (50 mM), as depicted in *Figure 7A*, pHluorin fluorescence increased by 356.8 ± 12.3% in the control condition and 352.3 ± 14.0% in the presence of AM251 (p=0.811, Student's *t*-test) (*Figure 7F and G*), indicating that the total amount of vesicles at individual boutons is not affected by GPR55 activation. It should be noted that $\Delta F_{400APs}$ by itself exhibited a large variability across boutons, showing a linear relation to the 2/3rd power (for calibration between volume and surface area of boutons) of that by $NH_4Cl$ ($\Delta F_{NH4Cl}$) (*Figure 7H*). Thus, the fraction of releasable vesicles in response to APs seemed constant among distinct boutons: larger boutons containing more vesicles simply show higher $\Delta F_{400APs}$. In addition, the predicted distribution pattern of $\Delta F_{400APs}$ with GPR55 activation calculated from the relation between the extent of GPR55-mediated release reduction and the $\Delta F_{400APs}$ (shown in *Figure 7E*) nicely overlapped the actual data obtained after the AM251 application (*Figure 7H*). Thus, the fluorescent live imaging of AP-triggered vesicle fusion revealed that GPR55 reduces vesicular fusion in response to APs, without changing total vesicles present at individual PC boutons.

To obtain further insights into the dynamics of synaptic vesicular pools possibly affected by GPR55, such as transitions among the readily releasable, recycling, reserve, and release-reluctant pools at individual boutons (*Fernandez-Alfonso and Ryan, 2008*; *Kim and Ryan, 2010*; *Cazares et al., 2016*; *Mori et al., 2021*), we repeatedly induced the 400 APs firing at the soma and measured the size of the fluorescence increase at boutons before and after the suppression of re-acidification of endocytosed vesicles with bafilomycin (100 nM), an inhibitor of vesicular ATPase (*Sankaranarayanan and Ryan, 2001*). As shown in *Figure 8A* (black trace), bafilomycin application enhanced the $\Delta F_{400APs}$ (by 55.9 ± 12.8%, p<0.001, paired *t*-test), and abolished the fluorescence decay after the end of the stimulation (relative $\Delta F_{400APs}$ remaining at 40 s after the end of APs: 59.8±18.6 [control] to 97.6 ± 4.6% [bafilomycin], p=0.0261, paired *t*-test), reflecting the inhibition of the re-acidification of endocytosed vesicles, which takes place in naïve conditions during and tens of seconds after the stimulation. Repeating the 400 APs stimulations after the bafilomycin application resulted in additive fluorescence increase, eventually reaching a plateau at 27.6 ± 2.6% of the maximal fluorescence increase observed after the $NH_4Cl$ application (*Figure 8A*). Imaging of presynaptic $Ca^{2+}$ increase with GCaMP7f expressed in PCs by AAV showed that $Ca^{2+}$ increase upon 400 APs remained constant during repetitive AP trains (*Figure 8—figure supplement 1A and B*). Thus, synaptic vesicles corresponding to those

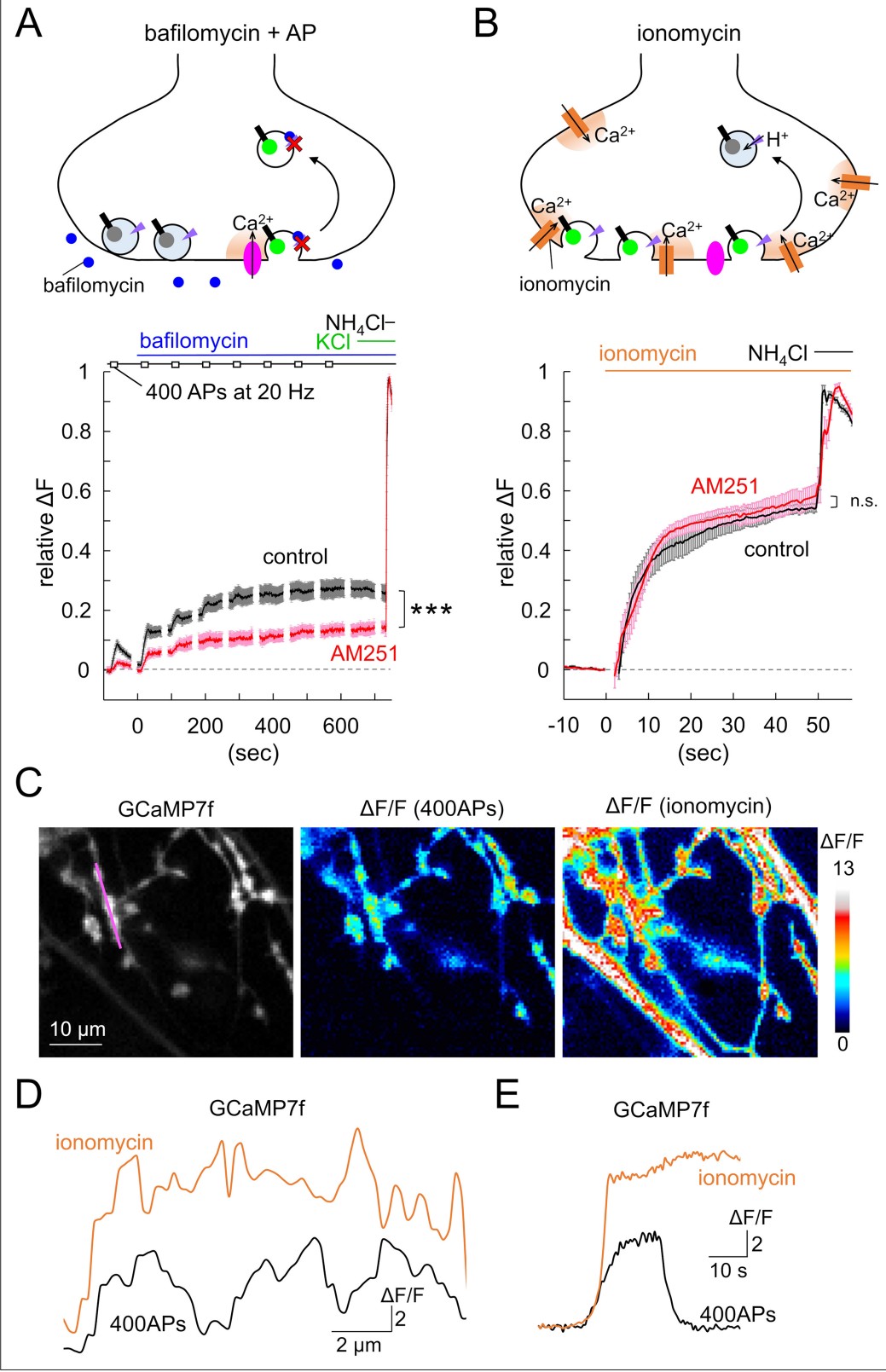

**Figure 8.** Increase in reluctant vesicles insensitive to APs by GPR55. (**A, B**) Schematic illustration (top) and time courses of synapto-pHluorin fluorescence changes at PC axon varicosities upon the repetitive 400 APs trains (20 Hz) before and after the application of bafilomycin and KCl (finally 50 mM) (control, n=13 cells; AM251, n=7) (**A**), or ionomycin (control, n=10 cells; AM251, n=8) (**B**), followed by the NH$_4$Cl application, in the presence or

*Figure 8 continued on next page*

*Figure 8 continued*

absence of AM251. Data are mean ± SEM. \*\*\*p<0.001; n.s., not significant. (**C**) Fluorescent images of GCaMP7f expressed in PC axon terminals (left) and color-coded relative fluorescence increase (ΔF/F) upon 400 APs at 20 Hz (middle) or the following ionomycin application (right). (**D, E**) Distribution patterns along an axonal segment (**D**; indicated as the magenta line in **C**) or representative time courses (**E**) of ΔF/F caused by 400 APs or ionomycin.

The online version of this article includes the following source data and figure supplement(s) for figure 8:

**Source data 1.** Data for pHluorin and GCaMP7f fluorescence shown in *Figure 8* and *Figure 8—figure supplement 1*.

**Figure supplement 1.** $Ca^{2+}$ influx induced by AP trains, external high $K^+$ or ionomycin.

for ~1000 APs are totally release-competent at individual axon terminals, in accord with previous studies (*Kawaguchi and Sakaba, 2015*). In order to induce the exocytosis of the remaining releasable vesicles in the terminal, if any, after the eight sets of 400 APs train stimulation, KCl (50 mM) was added to the external bath to potently depolarize the plasma membrane, resulting in widespread elevation of intracellular $Ca^{2+}$ (*Figure 8—figure supplement 1*). Nevertheless, little further increase in the pHluorin signal was observed upon KCl application (*Figure 8A*), confirming that most depolarization-sensitive vesicles have undergone fusion during the AP trains.

Intriguingly, the $\Delta F_{400APs}$ reduction by AM251 (red trace in *Figure 8A*) was evident throughout the 400 APs train stimulations (~10 min in total), reaching a plateau of about 13.2 ± 1.4% of the total amount of vesicles (*Figure 8A*, p<0.001, compared with control [27.6 ± 2.6%], Student's *t*-test). This implies that the total releasable vesicles upon repeated APs for minutes, which are composed of the RRP, the recycling pool, and even some reserve pool of vesicles, get smaller in number when GPR55 is activated. To further obtain insight into the reduced releasable vesicles by GPR55, we triggered vesicular release from PC boutons by causing an aberrant cytoplasmic $Ca^{2+}$ increase by adding ionomycin, which forms $Ca^{2+}$-conducting pores on the plasma membrane. Indeed, when monitored by GCaMP7f fluorescence imaging, ionomycin (10 μM) induced a much more potent and broader $Ca^{2+}$ increase throughout PC axons than the $Ca^{2+}$ increase induced by the inflow through VGCCs upon APs stimulation (*Figure 8C–E*, *Figure 8—figure supplement 1C*; ΔF/F upon 400 APs and ionomycin: varicosity, 5.3±0.4 and 16.2±0.9, p<0.001, Tukey–Kramer test; axon, 1.6±0.2 and 15.1±1.0, p<0.001, respectively). Consequently, surprisingly, application of ionomycin increased pHluorin signal to almost identical levels irrespective of the GPR55 activation (54.0 ± 1.6% and 58.1 ± 3.9% of total vesicles without and with AM251, respectively, p=0.356, Student's *t*-test) (*Figure 8B*). It should be noted that the size of the vesicle pool which underwent fusion upon the ionomycin application was almost double of the pool which underwent exocytosis upon thousands of APs. Thus, only a half population of $Ca^{2+}$-responsive releasable vesicles may be sensitive to APs in naïve PC axonal terminals, which is further reduced when GPR55 is activated.

## Discussion

In this study, we used a combination of subcellular patch-clamp recordings from axon terminals and fluorescence imaging of presynaptic vesicular exocytosis in cerebellar primary cultures, together with electrophysiology in cerebellar slices, and demonstrated that synaptic transmission from PCs is suppressed by activation of GPR55, a type of cannabinoid receptor. The mechanism of action involves a transformation of the presynaptic vesicles into insensitive to APs, while the $Ca^{2+}$ influx is kept constant in the terminal. Thus, a unique mechanism to negatively control synaptic outputs is unveiled here, which would provide a way to fine-tune the neuronal computation in the brain in coordination with the modulation of synaptic efficacy by conventional cannabinoid receptors.

### Modulation of synaptic function by endocannabinoids

Following the findings of critical roles of CB1 receptor in fine-tuning of neuronal circuit function, another type of cannabinoid receptor, GPR55, was identified (*Baker et al., 2006*; *Oka et al., 2007*). GPR55 seems to be activated by 2-AG and anandamide, similarly to CB1 or cannabinoid receptor type 2 (CB2), although the downstream signaling is different and also shows affinity to cannabidiol, another molecule found in cannabis (*Henstridge et al., 2008*; *Oka et al., 2009*). Distinct from typical chemical components of cannabis such as tetrahydrocannabinol (THC), cannabidiol has been shown

to modulate the function of the nervous system with limited addictive power, and thus has attracted attention for possible therapeutic applications to mental disorders (*Denis Völker et al., 2024*). Thus, the physiological actions of GPR55 are now increasingly studied, although its detailed effect on neuronal computation still remains largely veiled. Here, we observed a remarkable suppression of presynaptic function by GPR55 in PC boutons through a change in the releasable vesicle amounts and/or properties. Pioneering studies clarified an important role of GPR55 in synaptic transmission at hippocampal excitatory synapses, demonstrating presynaptic enhancement of glutamate release presumably by elevating the cytoplasmic residual $Ca^{2+}$ via release from intracellular stores (*Sylantyev et al., 2013*; *Rosenberg et al., 2023*), in contrast to the suppression of release in our observation. The lack of positive modulation of AP-triggered release through residual $Ca^{2+}$ in PC terminals might be due to abundant amounts of potent $Ca^{2+}$ buffer calbindin (*Fierro and Llano, 1996*). Indeed, increased vesicular fusion only for the AP-insensitive spontaneous vesicular release (as mIPSCs) was observed upon the $IP_3$-mediated $Ca^{2+}$ release from internal store (*Gomez et al., 2020*). Thus, minimal sensitivity of AP-triggered release to residual $Ca^{2+}$ in PC boutons would underlie the distinct effects of GPR55 activation at the presynaptic side. In addition, *Rosenberg et al., 2023* also demonstrated postsynaptic modulation by GPR55 at hippocampal inhibitory synapses: reduction of postsynaptic $GABA_A$ receptors, whereas limited presynaptic expression of GPR55 at axons. Here we showed clear GPR55 signals at PC terminals (see *Figure 2*), whereas no detectable changes in mIPSC amplitudes were observed upon GPR55 activation in potential DCN neurons (see *Figure 4E and F*). Thus, distinct combinations of critical factors, such as amount and/or location of GPR55 expression, subtypes of postsynaptic $GABA_A$ receptors, and/or their anchoring mechanism, might give rise to various forms of modulation of synaptic strength by GPR55, as demonstrated in the hippocampus and cerebellum.

One extensively studied activity-dependent short-term depression of synaptic strength in the CNS is DSE and/or DSI, which are typically caused by a strong postsynaptic depolarization accompanied with potent $Ca^{2+}$ increases or activation of $G_q$-coupled metabotropic receptors such as group-I mGluRs, activating PLC and/or diacylglycerol lipase. Both enzymes produce typical endocannabinoids such as 2-AG and anandamide, which then reach retrogradely presynaptic boutons beyond plasma membranes and synaptic cleft (*Kano et al., 2009*; *Castillo et al., 2012*). As a result, the CB1 receptor located at the presynaptic membrane is activated, and then $Ca^{2+}$-triggered transmitter release is suppressed for tens of seconds (*Kreitzer and Regehr, 2001*; *Ohno-Shosaku et al., 2001*; *Wilson and Nicoll, 2001*; *Diana et al., 2002*). Such short-term forms of presynaptic plasticity have been found to take place at a variety of synapses in the CNS including the hippocampus, cerebellum, prefrontal cortex, amygdala, and so on *Kano et al., 2009*; *Di Marzo et al., 2001*; *Katona et al., 2001*; *Hill and Patel, 2013*, which has provided basic knowledge about the primary mechanisms of cannabis intake affecting brain computation. In addition, the CB1 receptor has been shown to play roles in long-term synaptic plasticity and hence learning and memory in animals (*Kano et al., 2009*; *Gómez-Gonzalo et al., 2015*). On the other hand, the functional role of another cannabinoid receptor CB2 has been largely unknown. CB2 receptor exogenously expressed in hippocampal autaptic neurons from CB1 null mice was shown to suppress synaptic transmission and restore DSE (*Atwood et al., 2012*). In contrast, in spite of CB2 being expressed in PCs (*Ashton et al., 2006*), no modulation of synaptic outputs from PC axons has been demonstrated (*Hirono and Yanagawa, 2021*). Thus, as demonstrated here, PC axon terminals might be equipped with a compensatory mechanism to control outputs through GPR55 in place of CB2 receptor.

## Mechanisms for synaptic modulations by GPR55 distinct from CB1/CB2 receptors

Unlike typical metabotropic receptors, here we have shown that GPR55 reduces vesicular exocytosis by converting the $Ca^{2+}$-sensitive release-competent vesicles into reluctant ones upon AP arrival, without changing the $Ca^{2+}$ influx and vesicular $Ca^{2+}$ responsiveness (see *Figures 3–8*). Rapid replenishment and/or endocytosis of synaptic vesicles are known to depend on actin dynamics (*Lee et al., 2012*; *Miki et al., 2016*), which might be related to the altered vesicular sensitivity to $Ca^{2+}$ influx through VGCCs after the GPR55 activation. In any case, the unique control of presynaptic function by $G\alpha_{12/13}$-coupled GPR55 demonstrated here would provide a complementary modulation of synaptic outputs by cannabinoids together with CB1/CB2 receptors. Previous studies have clarified the molecular mechanisms by which CB1 receptor negatively modulates synaptic transmission. As well as a

typical inhibitory modulation by metabotropic receptors such as group II metabotropic glutamate receptors and GABA$_B$ receptors (*Kew et al., 2001*; *Takahashi et al., 1998*; *Bowery et al., 2002*), CB1 and CB2 receptors are thought to couple to G$_{i/o}$-type trimeric G-protein, negatively regulating the activation and/or conductance of VGCCs (*Guo and Ikeda, 2004*). Taking it into consideration that the vesicular exocytosis steeply depends on the local Ca$^{2+}$ concentration around the Ca$^{2+}$ sensor, synaptotagmin (*Schneggenburger and Neher, 2000*), the reduction of Ca$^{2+}$ influx is enough to explain the suppression of transmitter release. Interestingly, when activated for minutes in cerebellar granule cells, CB1 receptor was also shown to downregulate the release competence of vesicles through their displacement from active zone (*Ramírez-Franco et al., 2014*), which is reminiscent of the GPR55-mediated RRP reduction we presented in this study. It would be interesting to highlight the difference in mechanisms how distinct G-protein cascades modulate the release machinery in the future study.

## Dynamics of synaptic vesicles among distinct functional pools

Inspired by the GPR55-mediated marked decrease in RRP vesicles demonstrated here by the presynaptic C$_m$ measurement and biophysical analysis, we live-imaged vesicular fusion with synapto-pHluorin to functionally dissect the altered pools of vesicles in PC terminals (see *Figures 7 and 8*). Surprisingly, the reduction of release upon GPR55 activation was only evident when the release was triggered by APs, but undetectable in response to ionomycin, indicating that the apparent RRP reduction would be highly related to the source of cytoplasmic Ca$^{2+}$ increase. Taking into consideration that the RRP reduction was also evident when the Ca$^{2+}$ increase was triggered by much stronger stimulation (50 ms direct presynaptic depolarization or extracellular KCl application), the discrepancy between the effects of GPR55 on release by the presynaptic depolarization and the Ca$^{2+}$ ionophore could be reasonably ascribed to the relationship between release sites and Ca$^{2+}$ sources. Alternatively, slower kinetics and more prolonged duration, in addition to higher amplitude, of Ca$^{2+}$ elevation upon ionomycin might somehow have resulted in more efficient trigger of exocytosis. Notably, our imaging experiments indicated that the total recycling pool of vesicles which underwent exocytosis by thousands of APs during ~10 min corresponds to only ~25% of total vesicles, that is, a half of Ca$^{2+}$-sensitive release-competent vesicles that could fuse upon aberrant Ca$^{2+}$ increase via the Ca$^{2+}$-conducting ionophore (see *Figure 8A and B*). Because of the extremely potent and rapid Ca$^{2+}$ buffering system in the cytoplasm of PCs due to the abundant expression of calbindin (*Fierro and Llano, 1996*), the Ca$^{2+}$ influx from VGCCs is expected to activate vesicles with the Ca$^{2+}$ sensor very close to the VGCCs. Thus, it is implied that the release-competent vesicles relatively far from VGCCs tend to stay at their position for minutes without being exocytosed or getting into the AP-sensitive pools. Indeed, it was shown that the relatively large increase in the cytoplasmic Ca$^{2+}$ concentration by IP$_3$-caused release from the internal Ca$^{2+}$ store in PC terminals is only effective for the AP-independent spontaneous vesicular releases detected as miniature postsynaptic responses (*Gomez et al., 2020*). Thus, at least in PCs, the functional categorization of depolarization-sensitive vesicles seems to be influenced by the physical distance of vesicles and the source of Ca$^{2+}$. Interestingly, the reduction of releasable vesicles by GPR55 activation is limited to the AP-sensitive ones among Ca$^{2+}$-sensitive ones (see *Figure 8*). Taking the two findings together into consideration that GPR55 suppresses more potently the release caused by smaller Ca$^{2+}$ increase in the presence of high EGTA (see *Figure 5*) and delays the onset of release (see *Figure 6*), it would be reasonable to presume that the functional coupling between VGCCs and release machinery is loosened by the GPR55 activation. The loosened Ca$^{2+}$-release coupling, if any, upon GPR55 activation operating as the mechanism for synaptic suppression would compromise with the lack of changes in frequency of mIPSCs (see *Figure 4B and C*). The fact that GPR55 activation leads to altered actin cytoskeleton organization also makes this scenario plausible. Alternatively, it is also conceivable that GPR55 rather decreases the Ca$^{2+}$-dependent vesicle replenishment (*Hosoi et al., 2007*), which possibly leads to lower occupancy of release sites with release-competent vesicles, resulting in smaller total exocytic events in response to long depolarization (like that for 50 ms). This scenario would compromise with the tendency of more potent suppression of release by GPR55 with stronger Ca$^{2+}$ buffering by higher EGTA (see *Figure 5D*). On the other hand, it would also be possible that GPR55 somehow drives immature vesicles to the release sites, such as those immediately after endocytosed from the plasma membrane (*Tran et al., 2023*), leading to apparent reduction of transmitter release. Recent direct patch-clamp recordings combined with super-resolution fluorescence or electron microscopic imaging are highlighting the dynamic change of the Ca$^{2+}$-release

coupling distance and/or efficacy (*Fukaya et al., 2021*; *Midorikawa et al., 2024*). The exact change of release machinery relative to VGCCs at high resolution of ~nm in PC boutons upon GPR55 activation is a critical issue to be clarified in the future.

### Issues remain to be clarified in GPR55-mediated synaptic modulation

There are important issues to be unveiled in the future, as clarified for CB1 receptor: what neuronal situation leads to the production of endogenous GPR55 ligands like LPI, and how are the synapses modulated in terms of strength and time course? Multiple isoforms of LPI are known to have different chain length and unsaturation level, leading to distinct influence on GPR55 (*Brenneman et al., 2025*). The soy-derived LPI (obtained from Merck #L7635) which we used in this study, contains ~58% C16:0 and ~42% C18:0 or C18:2, and is known to exert neuroprotective and anti-inflammatory effects in cultured DRG neurons (*Brenneman et al., 2025*), together with just a little of the C20:4 isoform promoting neuroinflammatory action. Intracellular $Ca^{2+}$ elevation, such as pharmacologically caused seizures, leads to the LPI production at various CNS regions including the hippocampus (*Yamashita et al., 2010*; *Rosenberg et al., 2023*), although little is known about LPI in the cerebellum. Expression of LPI-producing phospholipase A1 and A2 in PCs (*Shirai and Ito, 2004*) would be a clue to study this issue.

In spite of the labeling of PC boutons with a fluorescent analogue of AM251 dependent on GPR55 gene expression (see *Figure 2*), direct evidence of specific localization of GPR55 at PC boutons, for example, by immuno-labeling, is missing at present due to the unavailability of a qualified and specific antibody as far as we have tested. Further, the present study could not identify the downstream mechanism how GPR55 deprives the release competent vesicles of sensitivity to APs. GPR55-coupled $G\alpha_{12/13}$–RhoA–actin pathway is expected to underlie, and detailed biophysical analysis in the future would answer which of $Ca^{2+}$-release coupling or $Ca^{2+}$-dependent vesicle replenishment is a likely candidate.

## Materials and methods

### Key resources table

| Reagent type (species) or resource | Designation | Source or reference | Identifiers | Additional information |
|---|---|---|---|---|
| Strain, strain background (*Rattus norvegicus*, either sex) | Wistar rat | Japan SLC, Inc. | Slc:Wistar | |
| Cell line (*Homo sapiens*) | AAVpro 293T Cell Line | Clontech | RRID:CVCL_B0XW | |
| Recombinant DNA reagent | pAAV-CA-WPRE | *Higashi et al., 2024* | https://doi.org/10.1126/sciadv.adj2547 | |
| Recombinant DNA reagent | pAAV-CA-EGFP | *Kawaguchi and Sakaba, 2015* | https://doi.org/10.1016%20/j.neuron.2015.02.013 | AAV vector 2 to transfect EGFP in neurons |
| Recombinant DNA reagent | pAAV-CA-synapto-pHluorin | *Kawaguchi and Sakaba, 2015* | https://doi.org/10.1016%20/j.neuron.2015.02.013 | AAV vector 2 to transfect synapto-pHluorin in neurons |
| Recombinant DNA reagent | pAAV-CA-jGCaMP7f | *Higashi et al., 2024* | https://doi.org/10.1126/sciadv.adj2547 | AAV vector 2 to transfect GCaMP7f in neurons |
| Recombinant DNA reagent | Tet-Off Advanced expression system | Clontech | Cat# 631070 | |
| Recombinant DNA reagent | pAAV-TRE-WPRE | This paper | | See 'Materials and methods' |
| Recombinant DNA reagent | pAAV-TRE-EGFP | This paper | | See 'Materials and methods' |
| Recombinant DNA reagent | pAAV-CMV-WPRE | This paper | | See 'Materials and methods' |
| Recombinant DNA reagent | pAAV-CMV-tTA | This paper | | See 'Materials and methods' |

*Continued on next page*

*Continued*

| Reagent type (species) or resource | Designation | Source or reference | Identifiers | Additional information |
|---|---|---|---|---|
| Recombinant DNA reagent | pSpCas9(BB)–2A-miRFP670 | Addgene | RRID:Addgene_91854 | |
| Sequence-based reagent | gRNAs specific to GPR55 | This paper | | Guide RNAs designed using CHOPCHOP; see 'Materials and methods' |
| Commercial assay or kit | AAVpro Helper Free System | Takara | Cat# 6230 | |
| Commercial assay or kit | AAVpro Purification Kit (All Serotypes) | Takara | Cat# 6666 | |
| Chemical compound, drug | AM251 | Tocris | Tocris: 1117/1 | |
| Chemical compound, drug | AM281 | Tocris | Tocris: 1115/10 | |
| Chemical compound, drug | Lysophosphatidylinositol | Sigma-Aldrich | Merck: 62966-1MG | |
| Chemical compound, drug | CID16020046 | Tocris | Tocris: 4959/5 | |
| Chemical compound, drug | Tocrifluor T1117 | Tocris | Tocris: 2540/100U | |
| Chemical compound, drug | Bafilomycin | Tocris | Tocris: 1334/100U | |
| Chemical compound, drug | Ionomycin | Wako | Wako: 095-05831 | |
| Chemical compound, drug | Tetrodotoxin | Nacalai Tesque | Nacalai: 32775-51 | |
| Chemical compound, drug | NBQX | Tocris | Tocris: 1044/10 | |
| Chemical compound, drug | Tetraethylammonium | Nacalai Tesque | Nacalai: 33013-62 | |
| Software, algorithm | ImageJ | NIH | RRID:SCR_003070 | |
| Software, algorithm | PatchMaster | HEKA Electronics | RRID:SCR_000034 | http://www.heka.com/downloads/downloads_main.html#down_patchmaster |
| Software, algorithm | Igor Pro | WaveMetrics | RRID:SCR_000325 | https://www.wavemetrics.com/ |
| Software, algorithm | Taro Tools | Labrigger, developed by Dr. Taro Ishikawa | https://labrigger.com/blog/2011/07/21/taro-tools-and-ppt-for-igor-pro/ | https://sites.google.com/site/tarotoolsregister/ |

## Animal usage

All experimental procedures were performed in accordance with regulations on animal experimentation in Kyoto University and approved by the local committee in Graduate School of Science, Kyoto University. Wistar rats (Slc: Wistar; Japan Slc Inc, Hamamatsu, Japan) of either sex were used in this study. Rats were housed with a mother rat until weaning in filter-top cages with bedding at 20–24°C under a 12/12 h light/dark photocycle. Food and water were available ad libitum. Rats were killed by decapitation, and older ones by anesthetization with isoflurane in air, then decapitation.

## Slice preparation

Acute sagittal slices (200 μm thickness) of cerebellum were prepared from Wistar rats at P28–42. Rats were anesthetized with isoflurane and perfused transcardially with ice-cold sucrose solution containing the following (in mM): NaCl 60, sucrose 120, $NaHCO_3$ 25, $NaH_2PO_4$ 1.25, KCl 2.5, D-glucose 25, ascorbic acid 0.4, myo-inositol 3, Na-pyruvate 2, $CaCl_2$ 0.1, $MgCl_2$ 3, pH 7.3–7.4, 300–350 mOsm/

kg $H_2O$ with continuous bubbling with mixed gas (95% $O_2$ and 5% $CO_2$). After decapitation, the cerebellum was quickly removed and cut with a Leica vibroslicer (VT1200S) in an ice-cold K-gluconate-based solution containing the following (in mM): K-gluconate 130, KCl 15, EGTA (ethylene glycol-bis(β-aminoethyl ether)-N,N,N′,N′-tetraacetic acid) 0.05, HEPES (4-(2-hydroxyethyl)–1-piperazineetha nesulfonic acid) 20, glucose 25, and D-AP5 50 µM, pH 7.4, 300–350 mOsm/kg $H_2O$, with continuous bubbling with mixed gas (95% $O_2$ and 5% $CO_2$). Slices were then incubated at 37°C for a half to 1 h in an extracellular solution containing the following (in mM): NaCl 125, $NaHCO_3$ 25, $NaH_2PO_4$ 1.25, KCl 2.5, D-glucose 25, ascorbic acid 0.4, myo-inositol 3, Na-pyruvate 2, $CaCl_2$ 2, $MgCl_2$ 1, pH 7.3–7.4, 300–320 mOsm/kg $H_2O$ with continuous bubbling with mixed gas.

## Cerebellar primary cultures

The method for preparing primary dissociated cultures of cerebellar neurons was similar to that in previous studies (*Kawaguchi and Hirano, 2007*). Briefly, cerebella were dissected out from newborn rats and their meninges were removed. The cerebella were incubated in $Ca^{2+}$ and $Mg^{2+}$-free Hank's balanced salt solution containing 0.1% trypsin and 0.05% DNase for 15 min at 37°C. Cells were dissociated by trituration and seeded in Dulbecco's modified Eagle's medium: nutrient mixture F12-based medium containing 2% fetal bovine serum. On the next day, ~80% of medium was replaced by basal medium Eagle. After that, half of the medium was changed every 3–4 days. Cytosine arabinoside (4 µM) was added to the medium to inhibit proliferation of glial cells. PCs could be visually identified by large cell bodies and thick dendrites. Experiments were performed >21 days after preparation of the culture.

## AAV production

The AAVpro 293T cell line (#632273, Clontech), a derivative of HEK293T cells capable of efficient AAV production, was used for viral packaging. Prior to shipment, the manufacturer confirmed that the cell line lacks mycoplasma contamination. Cells were maintained in Dulbecco's modified Eagle's medium (Nacalai Tesque) supplemented with 10% fetal bovine serum (Nichirei Biosciences) and penicillin–streptomycin (0.1 mg/ml, Nacalai Tesque) at 37°C in a humidified atmosphere containing 5% $CO_2$. AAV serotype-2 vectors were generated using the AAVpro Helper Free System (AAV2, #6230, Takara) according to the manufacturer's instructions.

## DNA construction and transfection

Plasmids containing coding DNA sequences of EGFP, synapto-pHluorin, and GCaMP7f were the same as previous studies (*Kawaguchi and Sakaba, 2015*; *Higashi et al., 2024*). The coding sequences were inserted into AAV expression vectors under control of CA (pAAV-CA-WPRE; *Higashi et al., 2024*), TRE (pAAV-TRE-WPRE) or CMV promoters (pAAV-CMV-WPRE) that were obtained by insertion of PCR-amplified WPRE sequence at the BamHI site of AAV-CMV (Takara). TRE promoter was obtained by PCR from the Tet-Off Advanced system (Clontech), and inserted into the HindIII-EcoRI site of pAAV-CMV-WPRE. Likewise, tTA (EcoRI-BamHI fragment) of pTet-OFF Advanced vector was inserted to the pAAV-CMV-WPRE vector. PCs were transfected with an AAV vector serotype 2 at 4–7 days after seeding. For GPR55-KD, the *S. pyogenes* Cas9 vector plasmid, pSpCas9(BB)–2A-miRFP670 was obtained from Addgene (#91854). Two gRNAs specific to GPR55 were designed with the CHOP-CHOP online software. The target sequences were designed to be separated by 81 base pairs so that the extracellular domain for AM251 binding would be deleted (*Joberty et al., 2020*). The target sequences were cloned into the plasmid based on the published protocol by *Ran et al., 2013*. Briefly, pSpCas9(BB)–2A-miRFP670 was digested with BbsI (New England Biolabs, USA) and the following oligos, 5'-CACCGCCCCTGTGGAAAGTGTAGAT-3' and 5'-AAACATCTACACTTTCCACAGGGGC-3', or 5'-CACCGTGCGAGAGTCTTCTTCCCCC-3' and 5'-AAACGGGGGAAGAAGACTCTCGCAC-3' (Eurofins Genomics, Japan), were annealed and ligated into the digested plasmid. For GPR55-KD using CRISPR/Cas9 or some imaging experiments using synapto-pHluorin, the DNA expression plasmids encoding Cas9 and gRNAs, or synapto-pHluorin were directly injected into the nucleus of PCs through sharp glass pipettes (*Kawaguchi and Hirano, 2007*). Recordings from DNA-injected cells were performed 1–2 days after the injection, except for GPR55-KD cells with CRISPR/Cas9-mediated acute genome editing (at 5 days).

## Electrophysiology

Patch-clamp recording was performed with an amplifier (EPC10, HEKA Elektronik, Germany) at room temperature (20–24°C), in an extracellular solution mentioned above for slices, or that containing the following for culture (in mM): NaCl 145, HEPES 10, glucose 10, $CaCl_2$ 2, $MgCl_2$ 1, pH 7.3–7.4 adjusted with KOH, and 300–310 mOsm/kg $H_2O$, using an inverted IX71 microscope (Olympus, Tokyo, Japan) equipped with a 40×, 0.95 numerical aperture (NA) objective or an upright BX51WI (Olympus) equipped with a 60×, 1.0 NA objective. Images were obtained with a Zyla4.2 sCMOS camera (Andor; Oxford Instruments, Oxford, UK). In some experiments, 2,3-Dioxo-6-nitro-1,2,3,4-tetrahydrobenzo[f] quinoxaline-7-sulfonamide (NBQX, 10 μM), tetrodotoxin (TTX, 1 μM), and tetraethylammonium (TEA, 2 mM) were applied to the bath. For current-clamp recordings, KCl-based internal solution with the following composition (in mM) was used: KCl 147, HEPES 10, EGTA 0.5, Mg-ATP 2, Na-GTP 0.2, pH 7.3–7.4 adjusted with KOH, and 320–330 mOsm/kg $H_2O$. For voltage-clamp recordings from PC-target neurons, a patch pipette was filled with a CsCl-based internal solution (pH 7.3–7.4 adjusted with CsOH and 320–330 mOsm/kg $H_2O$) containing (in mM): CsCl 147, HEPES 10, EGTA 0.5, Mg-ATP 2, Na-GTP 0.2. For voltage-clamp recordings from PC axon terminals, the internal solution was composed of (in mM): CsCl 103, KCl 44, HEPES 10, EGTA 0.5, Mg-ATP 2, Na-GTP 0.2 (pH 7.3–7.4 adjusted with CsOH and 320–330 mOsm/kg $H_2O$). In a subset of experiments shown in *Figure 5*, the EGTA concentration was increased to 5 mM. Presynaptic terminal recordings were performed in the presence of external TTX (1 μM) and TEA (2 mM). AM251 (5 μM for slice or 500 nM for culture; $EC_{50}$=39 nM, *Ryberg et al., 2007*), LPI (1 μM; $EC_{50}$=30 nM, *Oka et al., 2007*), or CID16020046 (5 μM; $IC_{50}$=0.21 μM, *Kargl et al., 2013*) were applied to the bath. LPI derived from soy (Merck, catalog #L7635) that was estimated to contain 58% C16:0 and 42% C18:0 or C18:2 was applied to the bath. Membrane potential of a PC was held at –70 mV unless otherwise stated. PCs' target postsynaptic neurons were voltage clamped at –70 to –100 mV (to avoid unclamped $Na^+$ currents) in slices, or at –70 mV in culture for IPSC recordings. Series resistance in recordings from the PC soma and axon terminal was online-compensated by 30–60%. For IPSC measurements, the remaining resistance was corrected off-line after the recording. In paired whole-cell recordings from a presynaptic PC soma and its postsynaptic neuron, APs were elicited by voltage pulses to 0 mV for 1–5 ms into PC soma. In the slice, IPSCs were evoked by a glass pipette placed in the white matter. The deconvolution analysis was performed as in a previous study (*Kawaguchi and Sakaba, 2015*) using a template of mIPSC waveform. The onset of release was defined as the first time point at which the calculated vesicular fusion velocity showed change larger than 2 SDs of that at the basal condition. Average amplitude and frequency of mIPSCs for individual cells were calculated from ≥200 events showing amplitude (≥7 pA) and appropriate time course.

Measurement of $C_m$ was done using sine + DC technique (*Neher and Marty, 1982*) implemented on the PatchMaster software (HEKA Elektronik, Germany). Presynaptic terminals were held at –80 mV and the sine wave (1 kHz and the peak amplitude of 30 mV) was applied on the holding potential. Because membrane conductance fluctuates during the depolarizing pulse, $C_m$ was usually measured approximately 50 ms after the depolarization. To estimate the clamped area in a direct bouton recording, capacitive transients in response to a hyperpolarizing pulse (5 or 10 mV) were used. To normalize the variation in $Ca^{2+}$ currents and neurotransmitter release depending on the size of the bouton, presynaptic $Ca^{2+}$ current and $C_m$ were normalized by the $C_m$ under the voltage-clamp in each bouton (in *Figures 3–6*).

All electrophysiological data were obtained and analyzed with PatchMaster and Igor Pro (Wave-Metrics, Portland, OR, USA). APs and IPSCs were detected with a threshold detection algorithm (TaroTools; https://sites.google.com/site/tarotoolsregister/) implemented by Dr. Taro Ishikawa as Igor Pro extensions, and the selection of each event was visually confirmed.

## Fluorescence imaging

For GPR55 fluorescent labeling, cultured neurons were incubated with T1117 (20 nM) in extracellular solution for 2 min at room temperature. Following incubation, cells were washed once with fresh extracellular solution to remove unbound dye and immediately imaged with a confocal fluorescence imaging system FV1000 (Olympus) using a 60× 0.9 NA water-immersion objective (Olympus) equipped on an upright microscope BX61WI. Fluorescence imaging of pHluorin or GCaMP7f was performed using an upright microscope BX51WI (Olympus) through a 60× 1.0 NA objective (Olympus). Fluorescent excitation was delivered using an LED (M470L4, Thorlabs). Images of pHluorin or GCaMP7f

fluorescence in PC axon terminals were obtained at 0.5 Hz with Zyla4.2 sCMOS camera (Andor), and analyzed with SOLIS (Andor) or ImageJ (NIH). APs were evoked at the voltage-clamped PC soma by depolarizing pulses (0 mV, 5 ms). Bafilomycin (100 nM), ionomycin (10 μM), KCl (50 mM), and $NH_4Cl$ (50 mM) were applied to the bath.

## Statistics

Data are presented as mean ± SEM unless otherwise stated. Statistical significance was evaluated by paired *t*-test, unpaired Student's *t*-test for unpaired groups, one-way ANOVA, Dunnett's test, or Tukey–Kramer test for multiple comparisons. Statistical significance was defined as $p < 0.05$.

## Acknowledgements

We thank Drs. Mitsuharu Midorikawa and Federico Trigo for the critical reading of the manuscript and helpful comments. Japan Society for the Promotion of Science, KAKENHI grants 25K02362 (SK), 25H02611 (SK), 22H02721 (SK), 22K19360 (SK), 24K18217 (TI), and 21K15189 (TI). Takeda Science Foundation (SK). Naito Foundation (SK).

## Additional information

### Funding

| Funder | Grant reference number | Author |
| --- | --- | --- |
| Japan Society for the Promotion of Science | 25K02362 | Shin-ya Kawaguchi |
| Japan Society for the Promotion of Science | 25H02611 | Shin-ya Kawaguchi |
| Japan Society for the Promotion of Science | 22H02721 | Shin-ya Kawaguchi |
| Japan Society for the Promotion of Science | 22K19360 | Shin-ya Kawaguchi |
| Japan Society for the Promotion of Science | 24K18217 | Takuma Inoshita |
| Japan Society for the Promotion of Science | 21K15189 | Takuma Inoshita |
| Takeda Science Foundation | | Shin-ya Kawaguchi |
| Naito Foundation | | Shin-ya Kawaguchi |

The funders had no role in study design, data collection and interpretation, or the decision to submit the work for publication.

### Author contributions

Takuma Inoshita, Data curation, Formal analysis, Funding acquisition, Investigation, Visualization, Writing – original draft, Writing – review and editing; Shin-ya Kawaguchi, Conceptualization, Supervision, Funding acquisition, Validation, Investigation, Writing – original draft, Project administration, Writing – review and editing

### Author ORCIDs

Takuma Inoshita ⬤ https://orcid.org/0009-0008-5099-357X
Shin-ya Kawaguchi ⬤ https://orcid.org/0000-0002-8386-1185

### Ethics

This study was performed in accordance with the recommendations in the Guide for the Care and Use of Laboratory Animals of the National Institutes of Health, USA. All procedures were approved by the local committee for animal experiments in Graduate School of Science, Kyoto University (approval number 202509).

Reviewer #1 (Public review): https://doi.org/10.7554/eLife.105268.3.sa1
Reviewer #2 (Public review): https://doi.org/10.7554/eLife.105268.3.sa2
Reviewer #3 (Public review): https://doi.org/10.7554/eLife.105268.3.sa3
Author response https://doi.org/10.7554/eLife.105268.3.sa4

## Additional files

### Supplementary files
MDAR checklist

### Data availability
The source data supporting the findings of this study are available in the Kyoto University Research Information Repository (KURENAI): https://doi.org/10.57723/kds607406.

The following previously published dataset was used:

| Author(s) | Year | Dataset title | Dataset URL | Database and Identifier |
|---|---|---|---|---|
| Inoshita T, Kawaguchi S | 2026 | Data for: "Increased reluctant vesicles underlie synaptic depression by GPR55 in axon terminals of cerebellar Purkinje cells" | https://doi.org/10.57723/kds607406 | Kyoto University Research Information Repository (KURENAI), 10.57723/kds607406 |

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
