## [Editor Report · eLife Assessment]

This is an **important** study reporting that activation of the presynaptic GPR55 receptor suppresses synaptic transmission by modulating GABA release through the reduction of the readily releasable pool without affecting the presynaptic AP waveform and calcium influx. The evidence supporting this claim is **compelling** and based on an impressive array of techniques including patch-clamp recordings from the axon terminals of cerebellar Purkinje cells and fluorescent imaging of vesicular exocytosis. While the authors have strengthened their conclusions on several technical fronts in the revised version, further investigation is needed into the mechanism by which GPR55 activation might make vesicles insensitive to the rise in presynaptic [Ca²⁺] mediated by VGCCs, and the nature of the endogenous process that would activate this pathway in vivo.

---

## [Referee Report · Reviewer #1 (Public review)]

In this manuscript, the authors report that GPR55 activation in presynaptic terminals of Purkinje cells decrease GABA release at the PC-DCN synapse. The authors use an impressive array of techniques (including highly challenging presynaptic recordings) to show that GPR55 activation reduces the readily releasable pool of vesicle without affecting presynaptic AP waveform and presynaptic Ca2+ influx. This is an interesting study, which is seemingly well-executed and proposes a novel mechanism for the control of neurotransmitter release. However, the authors' main conclusions are heavily, if not solely, based on pharmacological agents that most often than not demonstrate affinity at multiple targets. Below are points that the authors should consider in a revised version.

Major points:

(1) There is no clear evidence that GPR55 is specifically expressed in presynaptic terminals at the PC-DCN synapse. The authors cited Ryberg 2007 and Wu 2013 in the introduction, mentioning that GPR55 is potentially expressed in PCs. Ryberg (2007) offers no such evidence, and the expression in PC suggested by Wu (2013) does not necessarily correlate with presynaptic expression. The authors should perform additional experiments to demonstrate presynaptic expression of GPR55 at PC-DCN synapse.

(2) The authors' conclusions rest heavily on pharmacological experiments, with compounds that are sometimes not selective for single targets. Genetic deletion of GPR55 would be a more appropriate control. The authors should also expand their experiments with occlusion experiments, showing if the effects of LPI are absent after AM251 or O-1602 treatment. In addition, the authors may want to consider AM281 as a CB1R antagonist without reported effects at GPR55.

(3) It is not clear how long the different drugs were applied, and at what time the recording were performed during or following drug application. It appears that GPR55 agonists can have transient effects (Sylantyev, 2013; Rosenberg, 2023), possibly due to receptor internalization. The timeline of drug application should be reported, where IPSC amplitude is shown as a function of time and drug application windows are illustrated.

(4) A previous investigation on the role of GPR55 in the control of neurotransmitter release is not cited nor discussed Sylantyev et al., (2013, PNAS, Cannabinoid- and lysophosphatidylinositol-sensitive receptor GPR55 boosts neurotransmitter release at central synapses). Similarities and differences should be discussed.

Minor point:

(1) What is the source of LPI? What isoform was used? The multiple isoforms of LPI have different affinities for GPR55.

Comments on revisions:

In this revised version, the authors have addressed my major concerns. Notably, they used CRISPR/Cas9 genetic knockdown of GPR55 to independently validate their original findings. The main conclusions are now well supported and represent an important contribution to the field.

---

## [Referee Report · Reviewer #2 (Public review)]

Summary:

This paper investigates the mode of action of GPR55, a relatively understudied type of cannabinoid receptors, in presynaptic terminals of Purkinje cells. The authors use demanding techniques of patch clamp recording of the terminals, sometimes coupled with another recording of the postsynaptic cell. They find a lower release probability of synaptic vesicles after activation of GPR55 receptors, while presynaptic voltage-dependent calcium currents are unaffected. They propose that the size of a specific pool of synaptic vesicles supplying release sites is decreased upon activation of GPR55 receptors.

Strengths:

The paper uses cutting edge techniques to shed light on a little studied, potentially important type of cannabinoid receptors. The results are clearly presented, and the conclusions are sound.

Weaknesses:

The nature of the vesicular pool that is modified following activation of GPR55 is not definitively characterized.

Comments on revisions:

The authors have done a good job in answering the criticisms of reviewers. Consequently, the revised version offers a substantial improvement over the first version.

---

## [Referee Report · Reviewer #3 (Public review)]

Inoshita and Kawaguchi investigated the effects of GPR55 activation on synaptic transmission in vitro. To address this question, they performed direct patch-clamp recordings from axon terminals of cerebellar Purkinje cells and fluorescent imaging of vesicular exocytosis utilizing synapto-pHluorin. They found that exogenous activation of GPR55 suppresses GABA release at Purkinje cell to deep cerebellar nuclei (PC-DCN) synapses by reducing the readily releasable pool (RRP) of vesicles. This mechanism may also operate at other synapses.

Strengths:

The main strength of this study lies in combining patch-clamp recordings from axon terminals with imaging of presynaptic vesicular exocytosis to reveal a novel mechanism by which activation of GPR55 suppresses inhibitory synaptic strength. The results strongly suggest that GPR55 activation reduces the RRP size without altering presynaptic calcium influx.

Weaknesses:

The study relies on the exogenous application of GPR55 agonists. It remains unclear whether endogenous ligands released by physiological or pathological processes would have similar effects. There is also little evidence that GPR55 is expressed in Purkinje cell axon boutons. This study would benefit from the use of GPR55 knockout (KO) mice. The downstream mechanism by which GPR55 mediates the suppression of GABA release remains unknown.

Comments on revisions:

The authors have addressed all my concerns effectively. I have no further comments and want to commend their comprehensive study.

---

## [Author Response]

The following is the authors’ response to the original reviews

**Public Reviews:**

**Reviewer #1 (Public review):**
In this manuscript, the authors report that GPR55 activation in presynaptic terminals of Purkinje cells decrease GABA release at the PC-DCN synapse. The authors use an impressive array of techniques (including highly challenging presynaptic recordings) to show that GPR55 activation reduces the readily releasable pool of vesicle without affecting presynaptic AP waveform and presynaptic Ca^2+^ influx. This is an interesting study, which is seemingly well-executed and proposes a novel mechanism for the control of neurotransmitter release. However, the authors' main conclusions are heavily, if not solely, based on pharmacological agents that most often than not demonstrate affinity at multiple targets. Below are points that the authors should consider in a revised version.

We are happy to hear the encouraging comments from this reviewer, and thank for pointing out the important issues including the previous study design depending only on pharmacological agents. To address these, we have performed additional experiments, as detailed below.

Major points:(1) There is no clear evidence that GPR55 is specifically expressed in presynaptic terminals at the PC-DCN synapse. The authors cited Ryberg 2007 and Wu 2013 in the introduction, mentioning that GPR55 is potentially expressed in PCs. Ryberg (2007) offers no such evidence, and the expression in PC suggested by Wu (2013) does not necessarily correlate with presynaptic expression. The authors should perform additional experiments to demonstrate the presynaptic expression of GPR55 at PC-DCN synapse.

We completely agree with the reviewer in that our previous manuscript lacked the reliable information regarding presynaptic expression of GPR55 at PC boutons.

To clarify the localization, we first tried immunostaining of GPR55 using commercially available antibodies, but unfortunately they did not provide clear labeling of neurons and also even in GPR55-transfected HEK cells (used as positive control). Thus, we gave up the direct immunostaining. Alternatively, we attempted to label PC axonal boutons by GPR55-targeting dye together with a complementary strategy based on gene knock-down. Specifically, we used T1117, a fluorescent derivative of AM251 which is a GPR55 ligand used in the manuscript, and clear fluorescent signals were evident at GFP-labeled PC terminals. Still, by itself it was not clear whether the labeling was mediated by association with GPR55. Therefore, we also attempted to specifically suppress gene expression of GPR55 using CRISPR/Cas9-mediated genome editing in PCs, based on acute DNA micro-injection of plasmids into nuclei of PCs to express gRNAs targeting GPR55 together with Cas9. As a result, 5 days after the knock-down, T1117 labeling at axon terminals was reduced by ~50% compared to Cas9-alone controls. All these data are now shown in new Figure 2, and explained in the text p5-6, lines 141-159. Further, the reduction of GPR55 expression abolished the AM251-mediated reduction of vesicular exocytosis, as shown in new Figure 3D, E.

Taken together, these results essentially convince our main conclusions by strongly suggesting that GPR55 is present at PC axon terminals, where it negatively regulates the exocytosis upon activation by AM251.

(2) The authors' conclusions rest heavily on pharmacological experiments, with compounds that are sometimes not selective for single targets. Genetic deletion of GPR55 would be a more appropriate control. The authors should also expand their experiments with occlusion experiments, showing if the effects of LPI are absent after AM251 or O-1602 treatment. In addition, the authors may want to consider AM281 as a CB1R antagonist without reported effects at GPR55.

We thank the reviewer for pointing out these important issues. First, as noted above to confirm the presence of GPR55 at axon terminals of PCs, we performed genetic deletion of GPR55 using CRISPR/Cas9 system. In PCs co-expressing Cas9 and two gRNAs targeting the ligand-binding domain of GPR55, AM251 failed to suppress the exocytosis at PC boutons, together with decreased T1117 labeling. Therefore, the idea that GPR55 negatively regulates transmitter release at PC boutons has now been strengthened. The new data is shown in Figure 3D and E, and explained in the text p6, lines 173-178.

As suggested, we also carried out the occlusion experiments with LPI and AM251. First, LPI similarly reduced the readily releasable pool (RRP) size as AM251 did. Then, applied together, LPI and AM251 did not further reduce the RRP size compared with the effect by either compound alone. Thus, LPI and AM251 seem to act through the same pathway, consistent with the idea for role of GPR55 activation. The data is shown in new Figure 5—figure supplement 1 and explained in the text, p7-8, lines 215-221.

Regarding another point suggested by the reviewer, we applied AM281 and observed no effect on transmission at the PC–target neuron synapses (shown in new Figure 1F and I; explained in the text p5, lines 117-123), indicating that the effect of AM251 is likely to be mediated by GPR55, but not by CB1R.

Taken together, our additional experiments based on genetic and pharmacological experiments have consolidated our conclusion that GPR55 suppresses the presynaptic neurotransmitter release in PC boutons.

(3) It is not clear how long the different drugs were applied, and at what time the recordings were performed during or following drug application. It appears that GPR55 agonists can have transient effects (Sylantyev, 2013; Rosenberg, 2023), possibly due to receptor internalization. The timeline of drug application should be reported, where IPSC amplitude is shown as a function of time and drug application windows are illustrated.

Thank you for suggesting the better presentation of data. Accordingly, we have re-organized figures showing time course of changes in IPSCs before and after the drug application (new Figure 1 and 4; p4, lines 94-97; p5, lines 110-115; p7, lines 193-197). The current data presentation clearly shows that the effect of AM251 becomes evident in a few minutes after application, and somehow reaches a saturated level.

(4) A previous investigation on the role of GPR55 in the control of neurotransmitter release is not cited nor discussed (Sylantyev et al., 2013, PNAS, Cannabinoid- and lysophosphatidylinositolsensitive receptor GPR55 boosts neurotransmitter release at central synapses). Similarities and differences should be discussed.

We are really sorry for failing to adequately discuss this important work in our previous manuscript, and deeply appreciate the reviewer for pointing this out. We have now cited and discussed the work by Sylantyev et al. (2013), in the text (p12, lines 380-389), as following:

‘Pioneering studies clarified an important role of GPR55 in synaptic transmission at hippocampal excitatory synapses, demonstrating presynaptic enhancement of glutamate release presumably by elevating the cytoplasmic residual Ca^2+^ via release from intracellular stores (Sylantyev et al., 2013; Rosenberg et al., 2023), in contrast to the suppression of release in our observation. The lack of positive modulation of AP-triggered release through residual Ca^2+^ in PC terminals might be due to abundant amount of potent Ca^2+^ buffer calbindin (Fierro and Llano, 1996). Indeed, increased vesicular fusion only for the AP-insensitive spontaneous vesicular release (as mIPSCs) was observed upon the IP_3_-mediated Ca^2+^ release from internal store (Gomez et al., 2020). Thus, minimal sensitivity of AP-triggered release to residual Ca^2+^ in PC boutons would underlie the distinct effects of GPR55 activation at the presynaptic side.’

Minor point:(1) What is the source of LPI? What isoform was used? The multiple isoforms of LPI have different affinities for GPR55.

Thank you for letting us know about the lack of important information in the previous manuscript. In our experiments, we used a soybean-derived LPI mixture containing approximately 58% C16:0 and 42% C18:0 or C18:2 species. According to Brenneman et al. (2025), these isoforms show moderate or strong effects in cultured DRG neurons, whereas the C20:4 isoform, reported to promote neuroinflammatory signaling, was contained only at very low levels. We have added this information to the revised manuscript and briefly discussed the influence of different LPI isoforms on the physiological outcomes of GPR55 activation (p5, lines 127-131; p15, lines 493-496).

**Reviewer #2 (Public review):**
Summary:This paper investigates the mode of action of GPR55, a relatively understudied type of cannabinoid receptor, in presynaptic terminals of Purkinje cells. The authors use demanding techniques of patch clamp recording of the terminals, sometimes coupled with another recording of the postsynaptic cell. They find a lower release probability of synaptic vesicles after activation of GPR55 receptors, while presynaptic voltage-dependent calcium currents are unaffected. They propose that the size of a specific pool of synaptic vesicles supplying release sites is decreased upon activation of GPR55 receptors.Strengths:The paper uses cutting-edge techniques to shed light on a little-studied, potentially important type of cannabinoid receptor. The results are clearly presented, and the conclusions are for the most part sound.

We feel very happy to see the positive comments from the reviewer.

Weaknesses:The nature of the vesicular pool that is modified following activation of GPR55 is not definitively characterized.

We agree with the reviewer in that our data cannot fully address the changes of vesicle pools caused by GPR55. As detailed in responses to comments in ‘Recommendations for the authors’ from the reviewer, we have added explanation and discussion in the main text of the revised manuscript.

**Reviewer #3 (Public review):**
Summary:Inoshita and Kawaguchi investigated the effects of GPR55 activation on synaptic transmission in vitro. To address this question, they performed direct patch-clamp recordings from axon terminals of cerebellar Purkinje cells and fluorescent imaging of vesicular exocytosis utilizing synaptopHluorin. They found that exogenous activation of GPR55 suppresses GABA release at Purkinje cell to deep cerebellar nuclei (PC-DCN) synapses by reducing the readily releasable pool (RRP) of vesicles. This mechanism may also operate at other synapses.Strengths:The main strength of this study lies in combining patch-clamp recordings from axon terminals with imaging of presynaptic vesicular exocytosis to reveal a novel mechanism by which activation of GPR55 suppresses inhibitory synaptic strength. The results strongly suggest that GPR55 activation reduces the RRP size without altering presynaptic calcium influx.

We thank the reviewer for giving the encouraging comments on our study.

Weaknesses:The study relies on the exogenous application of GPR55 agonists. It remains unclear whether endogenous ligands released due to physiological or pathological activities would have similar effects. There is no information regarding the time course of the agonist-induced suppression. There is also little evidence that GPR55 is expressed in Purkinje cells. This study would benefit from using GPR55 knockout (KO) mice. The downstream mechanism by which GPR55 mediates the suppression of GABA release remains unknown.

We thank the reviewer for pointing out all of these important issues to be ideally addressed. As detailed in the responses to comments in the ‘Recommendations for the authors’ from the reviewers, we have addressed most of these weak points, and also added careful discussion in the text about the open questions to be solved in the future study.

**Recommendations for the authors:**

**Reviewer #2 (Recommendations for the authors):**
This is a high-quality paper that reports novel and interesting results. The authors should consider one main critique, related to Figure 6, as well as a number of minor points.

We thank the reviewer for making very positive assessment of our study. We have carefully considered the main critique regarding presynaptic vesicle pools (related to previous Figure 6), as well as other points, and accordingly revised manuscript.

Main critique:In Figure 6, it is said that GPR55 locks SVs in a state that is insensitive to VGCCs, based on a series of experiments with synapto-pHluorin. This conclusion is open to several critiques:The authors' model is shown in the diagram of Figure 6A. In this scheme, it appears as if recycled SVs eventually re-acidify in spite of the presence of bafilomycin, and that they are directed to a location close to the plasma membrane, but away from VGCCs. In fact, there is no evidence that the effects of bafilomycin could be limited in time. And there is a lot of evidence indicating that recycled SVs move back to release sites, close to VGCCs.

We are so sorry for presenting misleading figure panel in the previous Figure 6A. As the reviewer says, the effect of bafilomycin should be expected to last for long, and then the endocytosed vesicles cannot be re-acidified. Now, in new Figure 8A, we have changed the panel for explanation about the experimental situation of vesicles in the presence of bafilomycin. Another insightful point, kindly suggested by the reviewer, regarding the quick recruitment of newly endocytosed vesicles to release sites, is highly related to the interpretation of our data, but is a different issue from the situation explained in new Figure 8A. To avoid confusion, the arrow drawn in the previous version indicating the endocytosed vesicle movement back to the docked situation has been omitted in the new panel, and this critical issue is now carefully discussed in terms of the mechanism of GPR55 action on the release machinery (p15, lines 480-482).

The saturation of the train-induced signals is interpreted as reflecting an exhaustion of SVs initially close to VGCCs or more generally, susceptible to being released following VGCC activation.In an alternative scenario, saturation occurs because AP trains, or KCl applications, become unable to activate VGCCs. This could occur either because long illumination causes photodamage of VGCCs, or because repeated activation of VGCCs leads to their inactivation. The latter explanation is possible in spite of a publication from the authors' laboratory describing the facilitation of presynaptic VGCCs following paired stimulations in this synapse (Diaz-Rojas et al., 2015).

We agree that it is an important control experiment to demonstrate that Ca^2+^ increase upon repetitive AP trains is intact even during or after the long photo-illumination for imaging. To test this possibility, we have performed additional fluorescent Ca^2+^ imaging at PC varicosities during individual 400-AP trains and also in response to 50 mM KCl following the series of AP trains. Now new data demonstrated that Ca^2+^ influx remains constant across all AP trains (shown in Figure 8— figure supplement 1), arguing against VGCC inactivation or photodamage as a major factor underlying the saturated signal increase in the synapto-pHluorin. We have added explanation regarding this issue in the text p11, lines 327-329.

The authors explain the larger effect of ionomycin compared with AP trains and KCl applications as reflecting a better capacity to increase the bulk calcium concentration. The above proposal for the inactivation of VGCCs offers an alternative explanation, in my view more likely.

As noted above, our newly added Ca^2+^ imaging data clearly showed that individual AP trains induced similar Ca^2+^ influxes during repetitive trials, in line with our original interpretation. In addition, the Ca^2+^ increase by KCl was shown to be more potent and broader in axon terminals and trunks. Nevertheless, the exocytic signal caused by ionomycin was clearly large, implying a critical effect of the source of Ca^2+^ influx in PC boutons. Therefore, we suppose that the marked effect of ionomycin on release reflects higher elevation of bulk Ca^2+^ in the cytoplasm arising from non-site selective Ca^2+^-ionophore (Figure 8—figure supplement 1, p11, lines 327-334; lines 342-349).

In yet another scenario, recycled SVs in bafilomycin retain their fluorescence since they do not reacidify, but they come back to release sites to undergo new rounds of exocytosis. The new exocytosis events do not increase the fluorescence since the pH in the vicinity of synapto-pHluorin does not change. NH4Cl would then increase the fluorescence by revealing SVs that had not undergone exocytosis-endocytosis cycles during AP trains or KCl exposure. In this last scenario, the GPR55-sensitive SV pool would be a specific sub-pool of SVs that can be recycled by repetitive 400 AP trains.

We deeply appreciate the reviewer for pointing out this important possibility. We completely agree that this scenario can also explain the pool which is sensitive to GPR55. Therefore, we have added explanation of this possibility in the text (p15, lines 474–482).

Figure 6F shows calcium imaging measurements of PC varicosities. Unfortunately, crucial measurements are missing. It would have been revealing to compare calcium rises for the first and the last of the 8 400-AP trains. And to compare calcium rises elicited by 60 mM KCl before and after the series of 8 400-AP trains.

This is an important control experiment. Therefore, we have performed additional Ca^2+^ imaging during the eight 400-AP trains and KCl application. The new results shown in the present Figure 8—figure supplement 1 clearly suggest that Ca^2+^ rises are comparable between the first and eighth trains, and that additional Ca^2+^ influx (which was large in amplitude and wide in area) could still be evoked by KCl after the eight trains. The experiments are explained in the text p11, lines 327336.

Minor points:(1) Introduction: The Introduction would benefit from a more substantial description of what is known about GPR55 and downstream signaling pathways. Right now, it is stated that GPR55 is 'potentially expressed in PCs': What are the arguments behind this statement? Also, the signaling pathway is discussed on p.12, much too late in the ms. Why not move this section to the Introduction?

We thank the reviewer for the helpful suggestion. As recommended, in the revised manuscript, we have changed the Introduction by moving the sentences from other sections, including speculation about the expression of GPR55 in Purkinje cells (Ryberg et al., 2007; Wu et al., 2013) (p3-4, lines 71-75) and downstream signaling pathways (Gα_q_/PLC/IP_3_/Ca^2+^ and Gα_13_/RhoA/ROCK) (p3, 63-68).

(2) Legend to Figures 1, 2, and 4: What is the EGTA concentration in these experiments?

As suggested, the EGTA concentrations (0.5 or 5 mM) used in the individual experiments have now been clearly indicated both in the figure legends and in the Methods section (p18, lines 585586).

(3) Fig. 3C: These experiments show that some SV pool is depleted by AM251. The authors state that this is the RRP, but other options are possible. In the calyx of Held, similar experiments are supposed to deplete not only the FRP (=RRP, presumably) but also the SRP.

We thank the reviewer for pointing out the important aspect related to category for vesicle pools. In PC boutons, the membrane capacitance increases in response to different duration of depolarization pulses in a manner fitted by a single exponential curve (see Figure 5C for example). Our previous study (Kawaguchi and Sakaba, 2015) noted that the vesicle pools corresponding to FRP and SRP may not be easy to distinguish in PCs, suggesting apparently single component. That’s the reason why we simply describe the component as RRP in the present manuscript. Still, as suggested, careful discussion about typical fast- and slow components would be helpful to interpret our present findings. Therefore in the revised manuscript, we have added a sentence to explain this issue (p7, lines 211-214).

(4) p. 8: When the 400 APs protocol is introduced, the corresponding frequency (20 Hz?) should be mentioned. This information comes only much later in the ms.

We are sorry for our insufficient explanation in the previous manuscript. As suggested, we have clearly written the stimulation frequency ‘20 Hz’ in the main text where the 400 APs protocol first appears (p9, lines 277-278).

(5) Figure 5, panels B and F: synapto-pHluorin is labelled twice 'synapto-pHluolin'.

Sorry for careless typos. Now, those are corrected (new Figure 7).

(6) Legend to Figure 5, last line: 'x' is missing in the last equation.

Thank you for the careful and kind check. Now, ‘x’ has been added to the last equation in the legend for new Figure 7.

(7) p. 7, Interpretation of EGTA effects: The authors frame their interpretation of EGTA effects around the distance between release sites and VGCCs. However since AM251 appears to alter the recruitment of SVs, a more parsimonious interpretation would be that EGTA modifies the calciumdependent movement of SVs towards release sites.

Thank you for suggesting an insightful scenario. We agree that the capacitance jump upon long depolarization pulse would include exocytosis of substantial amount of vesicles which are newly recruited during the Ca^2+^ increase. Then, as the reviewer states, EGTA possibly lowers the Ca^2+^dependent replenishment of synaptic vesicles, and this replenishment system might be the target of GPR55 activation. Therefore, we have now clearly added an explanation about this possibility in the text (p15, lines 474-482).

(8) p. 13, Interpretation of GPR55 sensitive SV pool: The authors suggest a larger distance to VGCCs for this pool compared to naïve SVs. An alternative could be that in the presence of GPR55, the recruitment to release sites would be less efficient.

This is also an insightful suggestion to speculate the causal relationship between the GPR55mediated reduction of vesicular release and the vesicle pools. Accordingly, we have revised the Discussion (see “Dynamics of synaptic vesicles among distinct functional pools”) by clearly telling about the possibility of decreased recruitment of vesicles to release sites after the GPR55 activation (p15, lines 474-482). By totally considering all the suggested scenario, we believe that the possible mechanisms for GPR55-mediated reduction of release are much more clearly explained in the revised manuscript.

**Reviewer #3 (Recommendations for the authors):**
(1) The time course of the agonist-induced suppression should be reported (Figure 1).

This is an important point to show data clearly, as suggested also by the reviewer 1. Accordingly, we have changed the figure panels to show time courses of agonist-induced suppression (shown in new Figures 1 and 4).

(2) Show that the suppression of GABAergic transmission mediated by AM251 and LPI is eliminated in GPR55 KO mice.

We appreciate the reviewer for putting us to try this important experiment. Owing to the suggestion, we attempted to knock-down the GPR55 expression using CRISPR/Cas9 in cultured Purkinje cells. To avoid potential developmental compensations, here we adopted the CRISPR/Cas9-based genome editing approach, rather than using global knock out mice. Those GPR55-KO cells, as noted above in response to the comment #2 of reviewer #1, showed decreased fluorescent labeling of PC axon terminals to fluorescent-variant of AM251 (shown in new Figure 2) and abolishment of AM251-mediated suppression of vesicle exocytosis (Figure 3D and E). These results are explained in the text p5-6, lines 141-159; p6, lines 173-178.

(3) Include references supporting AM251 and LPI as GPR55 agonists and specify the E50 concentrations for each agonist. Furthermore, provide details about the GPR55 antagonist CID16600046.

As suggested, we have added references regarding GPR55 agonists, AM251 and LPI. In the text, the following information was added: AM251, originally characterized as an inverse agonist for CB1, has also been reported to act as a GPR55 agonist (Ryberg et al., 2007; Henstridge et al., 2009) (p5, lines 115-116). LPI is an established endogenous GPR55 agonist (Oka et al., 2007; Henstridge et al., 2009) (p5, lines 127-129). The reported EC_50_ values are ~ 30 nM for LPI (Oka et al., 2007, HEK cell assay) and 39 nM for AM251 (Ryberg et al., 2007, HEK cell assay) (p4, lines 94-95; p5, lines 127-129). Regarding the GPR55 antagonist CID16020046, detailed information (IC_50_ = 0.21 µM for GPR55 without significant effect on CB1 receptor) was added in the text with an appropriate citation (Kargl et al., 2013) (p5, lines 123-127). These points have also been added to the Methods section (p17, lines 587-589).

(4) Regarding the onset delay (Figure 4C; page 8, lines 3-4), consider the following: "AM251 induced a modest yet significant synaptic delay, estimated by the time to the onset of release" (or something similar).

We thank the reviewer for suggesting helpful explanation. Accordingly, we have changed the sentence to explain the delayed onset (p9, lines 264-265).

These three points should be properly acknowledged in the Discussion:(1) Are action potentials (APs)/depolarizations and ionomycin applications comparable? Ionomycin mediates a large calcium rise significantly slower than the calcium rise mediated by fast depolarization. Such presynaptic calcium dynamics could account, in part, for the different results.

The qualitative difference of Ca^2+^ increase between APs/depolarization-mediated ones and ionomycin-mediated one is an important point. Thank you for pointing out this issue. In the revised manuscript, we have added an explanation about the possible difference arising from the distinct dynamics of Ca^2+^ increases caused by direct depolarization of axon terminals or by ionomycin (p14, lines 452-453).

(2) Previous studies on hippocampal CA3-CA1 pyramidal cell synapses indicate that GPR55 activation enhances glutamate release through presynaptic calcium modulation while diminishing inhibitory postsynaptic strength by reducing GABAA receptors (Sylantyev et al., PNAS 2013; Rosenberg et al., Neuron 2023). In contrast, Inoshita and Kawaguchi discovered that GPR35 suppresses PC-DCN inhibitory transmission by decreasing GABA release without affecting inhibitory postsynaptic strength. Some potential explanation for this discrepancy is warranted.

We appreciate the reviewer for pointing out this important issue, and feel sorry for not providing an appropriate discussion about the possible interpretation in the previous manuscript. In the revised manuscript, we have added explanations for this discrepancy. First, PC terminals show only limited influence by elevated cytoplasmic Ca^2+^ through ER store on GABA release (Gomez et al., 2020) probably due to abundant calbindin. Second, our present data clearly show the GPR55 signals at PC terminals (although indirect, see Figure 2), while hippocampal inhibitory neuronal boutons somehow showed lower GPR55 levels compared with excitatory neuronal boutons (Rosenberg et al., Neuron, 2023). Third, the subtypes and/or anchoring mechanism for postsynaptic GABA_A_ receptors might be different between two distinct postsynaptic neurons in the hippocampus and the cerebellum. These factors are now clearly discussed in the text (p12, lines 380-396).

(3) Earlier work has suggested that CB1 receptor activation can alter the release machinery. Therefore, the observation that GPR55 activation induces changes in the RRP is not entirely surprising.

As pointed out, previous studies showed that CB1R influences the synaptic release machinery, rather than Ca^2+^ influx (Ramirez-Franco et al., 2014). In that context, as the reviewer says, the GPR55-mediated RRP change can be regarded as a similar synaptic modulation mechanism as the CB1-mediated one. However, considering the different downstream signaling pathways, G_12/13_- or G_q_-mediated one and G_i/o_-mediated one, our findings would provide an important scope about the regulation mechanisms of release machinery, which should be further analyzed in the future study. Now we have added these points in discussion (p13-14, lines 435-439).

(4) Add a section about the limitations of this study (see Weaknesses above).

As suggested, we have added a section about the limitations of this study at present, which we could not address in the revision and should be addressed in the future (p15, lines 488-508). Particularly, the actual endogenous agonist to activate GPR55, and the physiological situation in which the agonist is produced, much more direct evidence for GPR55 presence at PC boutons, and the downstream mechanisms of GPR55-mediated suppression of GABA release are now clearly notified in that section.

(5) Double-check grammar and typos ("anandamid").

We are really sorry for the poor writings in the previous manuscript. Now, we have carefully checked the text.